# Cyclin F drives proliferation through SCF-dependent degradation of the retinoblastoma-like tumor suppressor p130/RBL2

Taylor P Enrico[1,2], Wayne Stallaert[3], Elizaveta T Wick[1,2], Peter Ngoi[4], Xianxi Wang[2], Seth M Rubin[4], Nicholas G Brown[1,2], Jeremy E Purvis[2,3], Michael J Emanuele[1,2]*

[1]Department of Pharmacology. The University of North Carolina at Chapel Hill, Chapel Hill, United States; [2]Lineberger Comprehensive Cancer Center. The University of North Carolina at Chapel Hill, Chapel Hill, United States; [3]Department of Genetics. The University of North Carolina at Chapel Hill, Chapel Hill, United States; [4]Department of Chemistry and Biochemistry. University of California at Santa Cruz, Santa Cruz, United States

**Abstract** Cell cycle gene expression programs fuel proliferation and are universally dysregulated in cancer. The retinoblastoma (RB)-family of proteins, RB1, RBL1/p107, and RBL2/p130, coordinately represses cell cycle gene expression, inhibiting proliferation, and suppressing tumorigenesis. Phosphorylation of RB-family proteins by cyclin-dependent kinases is firmly established. Like phosphorylation, ubiquitination is essential to cell cycle control, and numerous proliferative regulators, tumor suppressors, and oncoproteins are ubiquitinated. However, little is known about the role of ubiquitin signaling in controlling RB-family proteins. A systems genetics analysis of CRISPR/Cas9 screens suggested the potential regulation of the RB-network by cyclin F, a substrate recognition receptor for the SCF family of E3 ligases. We demonstrate that RBL2/p130 is a direct substrate of SCF^cyclin F. We map a cyclin F regulatory site to a flexible linker in the p130 pocket domain, and show that this site mediates binding, stability, and ubiquitination. Expression of a mutant version of p130, which cannot be ubiquitinated, severely impaired proliferative capacity and cell cycle progression. Consistently, we observed reduced expression of cell cycle gene transcripts, as well a reduced abundance of cell cycle proteins, analyzed by quantitative, iterative immunofluorescent imaging. These data suggest a key role for SCF^cyclin F in the CDK-RB network and raise the possibility that aberrant p130 degradation could dysregulate the cell cycle in human cancers.

*For correspondence:
emanuele@email.unc.edu

## Editor's evaluation

The identification of the tumor suppressor RBL2/p130 as a substrate of cyclin F/SCF adds a new level of understanding about the role of this ubiquitin ligase in cell cycle control and identifies a novel functional interaction that could have implications for cancer. This work will be of interest to researchers in the fields of cell cycle and cancer biology.

## Introduction

The eukaryotic cell cycle consists of a sequential progression of events that govern cell growth and division. During cell cycle progression, many hundred genes oscillate in expression, contributing to

myriad processes, including DNA replication, chromosome segregation, cytoskeletal organization, and so on. The expression of cell cycle genes is repressed during quiescence, a reversible state of growth arrest, and in early G1-phase, presenting a critical barrier to proliferation. The retinoblastoma protein (RB) is a vital regulator of cell cycle gene repression. During quiescence and in early G1-phase, RB binds and inhibits E2F transcription factors, repressing transcription of many cell cycle genes. RB is phosphorylated by Cyclin-Dependent Kinases 4 and 6 (CDK4/6), as well as CDK2 (*Narasimha et al., 2014*; *Rubin et al., 2020*). This phosphorylation causes RB to dissociate from E2F, promoting the E2F-dependent expression of cell cycle genes that catalyze S-phase entry and cell cycle progression (*Rubin et al., 2020*). Due to its critical role in restricting proliferation, RB is a prototypical tumor suppressor (*Dyson, 2016*).

RB has two closely related family members, RBL1/p107 and RBL2/p130 (*Dyson, 1998*; *Sadasivam and DeCaprio, 2013*). Both p107 and p130 are also tumor suppressors that prevent cell cycle gene expression by binding the repressor E2F proteins E2F4/5 (*Claudio et al., 1994*; *Zhu et al., 1993*), and both are also regulated by CDK-dependent phosphorylation (*Canhoto et al., 2000*; *Farkas et al., 2002*; *Hansen et al., 2001*). Additionally, p130 functions as part of the mammalian DREAM complex (DP, RB-like, E2F4/5, and MuvB) (*Litovchick et al., 2007*; *Smith et al., 1996*). DREAM assembles during quiescence and inhibits cell cycle progression by restricting the transcription of numerous cell cycle genes regulated by E2F, B-MYB, and FoxM1 transcription factors (*Fischer et al., 2016*; *Müller et al., 2012*). Accordingly, perturbations to p130 or the DREAM complex allow expression of its cell cycle target genes, shifting the balance from quiescence toward proliferation (*Forristal et al., 2014*; *Iness et al., 2019*; *Patel et al., 2019*).

RB and p130 collaborate to suppress proliferation. In mice, *Rb⁻/⁻p130⁻/⁻* mouse embryo fibroblasts (MEFs) grow more rapidly in culture than MEFs deficient in either *Rb* or *p130* alone, and *Rb⁻/⁻p130⁻/⁻* mice spontaneously form many more tumors than their respective single-gene knockouts (*Dannenberg et al., 2004*). In mouse models of small cell lung cancer, *p130* knockout increases tumor size and overall tumor burden, even in the background of *Rb* and *p53* loss (*Ng et al., 2020*; *Schaffer et al., 2010*). Consistent with its role as a tumor suppressor, *p130* cooperates with RB to repress G2-M genes in response to genotoxic stress (*Schade et al., 2019*). And, *p130* loss in primary human fibroblasts leads to increased expression of cell cycle genes compared to loss of *Rb* alone (*Schade et al., 2020*). These observations highlight the importance of p130 in cell cycle control, as well as its role in tumor suppression. These results also illustrate the importance of the broader CDK-RB network in normal proliferation, and the consequence of its dysregulation in the aberrant cell cycles observed in cancer. RB mutations, overexpression of cyclin D and cyclin E, loss of p130 protein, and dysregulation of the mammalian DREAM complex have all been implicated in increased cellular proliferation and tumorigenesis (*Forristal et al., 2014*). Interestingly, p130 mutations are infrequent compared to other tumor suppressors like *RB, CDKN2A*, and *p53*, suggesting the possibility that post-translational mechanisms could account for its inactivation.

Cyclin F is a non-canonical cyclin—it neither binds nor activates CDKs (*Bai et al., 1994*; *D'Angiolella et al., 2013*). Instead, cyclin F is one of ~70 F-box proteins, a family of substrate recognition receptors that recruit substrates to the Skp1-Cul1-Fbox protein (SCF) E3 ligase (*Bai et al., 1996*; *Cardozo and Pagano, 2004*). SCF ligases play an evolutionarily conserved role in promoting cell cycle progression by triggering the destruction of cell cycle inhibitors. For example, yeast SCF^Cdc4 and human SCF^Skp2 trigger the destruction of CDK inhibitors Sic1 and p27, respectively (*Carrano et al., 1999*; *Feldman et al., 1997*; *Schwob et al., 1994*; *Skowyra et al., 1997*).

Cyclin F mRNA and protein levels oscillate during the cell cycle, giving cyclin F cell cycle-dependent activity (*Bai et al., 1994*). Cyclin F begins to accumulate at the G1/S transition, peaks in G2, and its protein levels are subsequently downregulated via proteasomal degradation in mitosis and G1 (*Choudhury et al., 2016*; *Mavrommati et al., 2018*) where cyclin F is ubiquitinated by SCF^βTRCP following cyclin F phosphorylation by casein kinase II (*Mavrommati et al., 2018*). Cyclin F is also ubiquitinated by the cell cycle E3 Anaphase Promoting Complex/Cyclosome (APC/C) (*Choudhury et al., 2016*). Further, cyclin F has been implicated in the ubiquitination and degradation of several cell cycle proteins (*Emanuele et al., 2020*). Taken together, its dynamic regulation and substrate repertoire highlight the importance of cyclin F in cell cycle control.

We demonstrate here that cyclin F binds, ubiquitinates, and regulates the RB-family tumor suppressor p130. We identify the regions in both cyclin F and p130 that are required for their

interaction. Interfering with cyclin F-p130 regulation, by expressing a mutant version of p130 which cannot be ubiquitinated, causes a severe defect in proliferation. Taken together, these data implicate cyclin F as a new, key player in CDK-RB network regulation, highlighting a critical ubiquitin-based mechanism that controls cell proliferation.

## Results

### Cyclin F/CCNF fitness correlates with the CDK-RB network

The Project Achilles Cancer Dependency Map (DepMap) has performed near genome-wide, CRISPR/Cas9 loss-of-function screens in hundreds of cancer cell lines. Fitness scores, corresponding to each gene in each cell line, reflect the relative impact of individual gene knockout on proliferation and survival. Despite differences in protocols and reagents, DepMap scoring is highly concordant with Project Score, another large, pan-cancer, CRISPR/Cas9 screening platform (*Dempster et al., 2019*). Importantly, systems genetics analyses of these data can be used to identify physically or genetically linked genes/proteins, since their loss often similarly impact overall fitness (*Munoz et al., 2016*; *Boyle et al., 2018*; *Doench et al., 2016*; *Kory et al., 2020*; *McDonald et al., 2017*; *Pan et al., 2018*; *Wang et al., 2017*). The Pearson's correlation coefficient was computed for DepMap fitness gene scores across cell lines (17,634 genes in 789 cell lines at the time of analysis). This analysis effectively detects pathways involved in proliferation. For example, the gene most highly correlated with the DNA helicase component MCM2 is its complex member MCM4 (*Figure 1—figure supplement 1A*). The genes most highly correlated with mitogen-activated protein kinase MEK1/MAP2K1 are its upstream activator BRAF and its downstream effector ERK2/MAPK1 (*Figure 1—figure supplement 1A*). Genes that encode proteins in the Skp2-Cyclin E/A-CDK2-E2F pathway are highly correlated, as are those involved in autophagy (*Figure 1—figure supplement 1A*).

Fitness scores for those genes in CDK-RB network which promote proliferation (e.g., *CDK4*, *CCND1/Cyclin D1, etc.*) are well correlated between DepMap and Project Score (*Figure 1—figure supplement 2*; *Dempster et al., 2019*). The gene most highly correlated with *CDK4* is its coactivator *CCND1/Cyclin D1,* demonstrating that relevant genetic relationships in the CDK-RB network are correlated (*Figure 1—figure supplement 3A*). Interestingly, the ninth most highly correlated gene with *CDK4* was *CCNF*, which encodes cyclin F (*Bai et al., 1994*; *Bassermann et al., 2014*). *CCNF* is among the top 1% of highly correlated genes when analyzing several other members of the CDK-RB network, including *CDK4, CDK6, CCND1, and RBL1* (*Figure 1—figure supplement 3A*). Accordingly, when we determined those genes most highly correlated with *CCNF* knockout, *CDK4, CDK6, CCND1/Cyclin D1, RBL1,* and *RBL2* are among the top 85 genes, out of more than 17,630 (~0.5%; *Figure 1A*). Thus, the impact of *CCNF* knockout on fitness is highly correlated with genes in the CDK-RB network.

Gene ontology (GO) analysis on the top 94 genes correlated with *CCNF* (Pearson correlation >0.15) enriched GO terms related to cell cycle, regulation of ubiquitin protein ligase activity, mitotic G1-phase and G1/S transition, Cyclin D1-CDK4-CDK6 complex, G1/S specific transcription, and Cyclin A-CDK2 associated events at S-phase entry (*Figure 1B*). When we filtered the top *CCNF* correlated genes by the GO term 'cell division' (*Figure 1C*), three sub-networks emerged: the Nde1-kinetochore complex, the Anaphase Promoting Complex/Cyclosome (APC/C), and the CDK-RB network. Identification of eight subunits of the APC/C complex is consistent with our previous observation of a reciprocal relationship between SCF^cyclin F and APC/C^Cdh1 (*Choudhury et al., 2016*). Among the remaining genes/proteins were two known cyclin F substrates: *E2F7*, which encodes the transcriptional repressor E2F7 (*Yuan et al., 2019*), and *FZR1*, which encodes the APC/C coactivator protein, Cdh1 (*Choudhury et al., 2016*).

We examined endogenous proteins corresponding to CDK-RB network genes that correlate most strongly with *CCNF* using two breast cancer cell lines, MCF-7 and T47D, engineered for doxycycline-inducible expression of cyclin F (*Wasserman et al., 2020*). Cells were treated with increasing amounts of doxycycline for 24 hr, and immunoblotted for CDK4, CDK6, cyclin D1, p107, and p130. We observed a specific downregulation of endogenous p130 in both cell lines, whereas all other proteins remained unchanged (*Figure 1D*). Similarly, p130 levels decrease in a time-dependent manner following doxycycline treatment, and this can be prevented by inhibiting the proteasome with MG132, or cullin RING E3 ligases, using the neddylation inhibitor MLN4924 (*Figure 1—figure supplement 3B*; *Ohh et al.,*

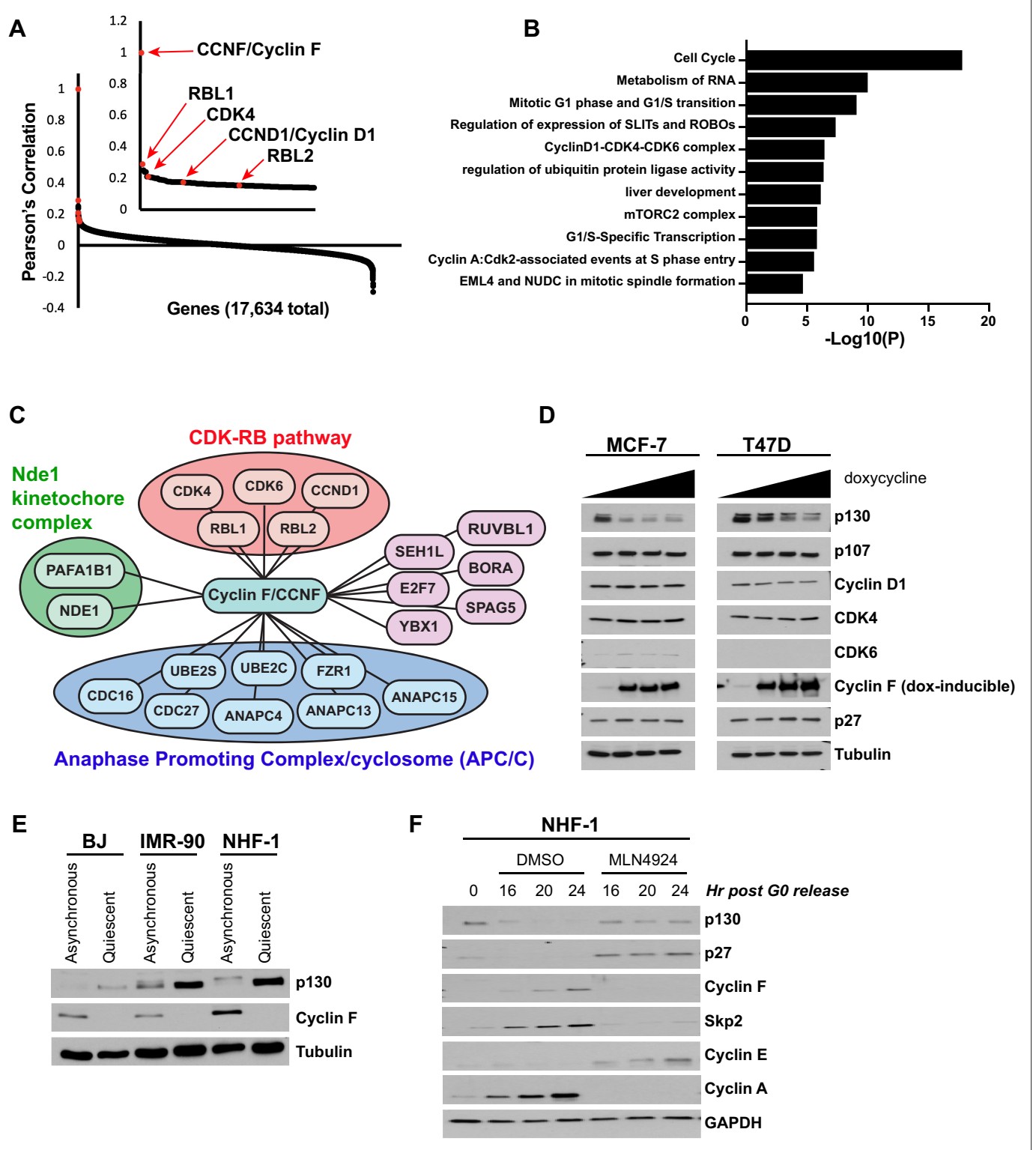

**Figure 1.** Analysis of the Cancer Dependency Map reveals that *CCNF* is highly correlated with the CDK-RB network. (**A**) Cancer Dependency Map data from project Achilles were analyzed to identify the impact of gene loss-of-function on cellular fitness, and fitness correlation with that of *CCNF,* based on pooled CRISPR/Cas9 gene knockout screens performed in 789 cell lines. Pearson's correlation coefficients are reported for all gene pairs (each dot corresponds to a single *CCNF*-gene X pair). The Pearson's correlations for *CCNF* compared to 17,634 other genes are shown. The CDK-RB network members highlighted in red all score in the top 0.5% of genes whose impact on fitness is most highly correlated with *CCNF*. (**B**) Gene ontology (GO) analysis was performed for the top 0.05% of genes whose impact on fitness is most highly correlated with *CCNF*. The top 10 enriched GO terms and

*Figure 1 continued on next page*

*Figure 1 continued*

their corresponding p-value is shown. (**C**) The top 0.05% of genes whose impact on fitness is most highly correlated with *CCNF* were sorted by the GO term cell division (GO:0051301). A graphical representation of the remaining 21 genes is shown. Genes are grouped by their known associations with specific functional pathways or complex, including CDK-RB, Nde1-kinetochore, or APC/C. (**D**) MCF-7 and T47D cells were engineered to contain a TET-inducible cyclin F transgene. Cells were treated with 0 (vehicle control), 5, 25, or 100 ng/ml of doxycycline to induce cyclin F expression, and the indicated proteins were analyzed by immunoblot. No band was detected for CDK6 in T47D cells. Representative of n=3 experiments. (**E**) BJ, IMR-90, and NHF-1 human fibroblast cell lines were synchronized in G0 by 48 hr serum starvation (quiescent) or allowed to proliferate normally (asynchronous). Whole-cell extracts were collected for immunoblot analysis. Representative of n=3 experiments. (**F**) NHF-1 cells were synchronized in G0 by serum starvation for 48 hr. Cells were then released into the cell cycle upon the addition of serum-containing media supplemented with either MLN4924 or vehicle (DMSO) as a control. Cells were collected at the indicated time points after release. Protein levels were assessed by immunoblot. Data represent n=3 independent experiments.

The online version of this article includes the following figure supplement(s) for figure 1:

**Figure supplement 1.** Analysis of Project Achilles Dependency Map reveals that highly correlated genes encode proteins known to interact.

**Figure supplement 2.** Gene dependency scores from the Broad Institute Project Achilles Dependency Map data set and the Sanger Project Score data set correlate highly.

**Figure supplement 3.** Analysis of Project Achilles Dependency Map reveals that *CDK-RB* network genes correlate highly with *CCNF*.

*2002*; *Soucy et al., 2009*). Taken together, these data suggest that cyclin F might regulate p130 degradation.

To further assess the relationship between cyclin F and p130, we compared asynchronous human fibroblasts to those same cells synchronized in quiescence by serum-starvation for 48 hr. In each of cell lines tested (BJ, IMR-90, and NHF-1), cyclin F protein levels significantly decrease in quiescence, whereas p130 protein levels increase, establishing an inverse expression pattern (*Figure 1E*). Next, NHF-1 cells were synchronized in quiescence by serum deprivation and then released into the cell cycle. Cyclin F protein began to accumulate as cells re-entered the cell cycle at G1/S, similar to what was previously reported (*Choudhury et al., 2017*; *D'Angiolella et al., 2012*; *Figure 1—figure supplement 3C*). Meanwhile, p130 was degraded as cells exited quiescence. The accumulation kinetics of cyclin F is similar to another F-box protein, Skp2, which ubiquitinates p27, marking it for degradation. Skp2 has also been implicated in p130 degradation (*Bhattacharya et al., 2003*; *Tedesco et al., 2002*). However, p130 has a different degradation pattern than p27, and in a previous study, p130 was still degraded in Skp2 knockout cells (*Tedesco et al., 2002*).

To determine whether the SCF and cullin E3 ligase family is required for p130 degradation at quiescence exit, we synchronized NHF-1 and T98G cells in quiescence and then released cells into the cell cycle in the presence of either DMSO (control) or MLN4924. In cells treated with MLN4924, neither p130 nor p27 is degraded, cyclin E accumulates but does not get degraded, and there is a significant delay in cyclin A accumulation (*Figure 1F* and *Figure 1—figure supplement 3D*).

To determine p130 levels in the absence of cyclin F, we first utilized *CCNF* knockout HeLa cells generated using CRISPR/Cas9 gene editing (*Choudhury et al., 2016*). In HeLa cells, p130 protein levels are higher in CCNF KO cells (sgCCNF) than in control cells (sgCTRL; *Figure 2A*). Next, CCNF KO and control HeLa cells were synchronized at G1/S by double thymidine block. After release, p130 levels were higher in the CCNF knockouts at every time point analyzed (*Figure 2—figure supplement 1A*). We then transiently depleted cyclin F from NHF-1 or IMR-90 cells. Cyclin F depletion increased p130 in both cell lines and with two different siRNAs (*Figure 2B*). Neither CCNF KO HeLa cells, nor the transiently depleted NHF-1 or IMR-90 cells, exhibited significant defects in overall cell cycle compared to their respective controls, judged by propidium iodide staining and analyzed by flow cytometry (*Figure 2—figure supplement 1B-D*). Consistent with the regulation of p130 by cyclin F resulting from a physical interaction, endogenous cyclin F co-immunoprecipitated endogenous p130 from HEK293T and U2OS cells (*Figure 2C*).

## p130 degradation is proteasome- and neddylation-dependent and requires the canonical cyclin F substrate-binding site

We determined if exogenously expressed cyclin F affects the stability of exogenously expressed p130 protein. In HEK293T cells transfected with HA-tagged p130 (HA-p130), increasing amounts of FLAG-tagged cyclin F (FLAG-cyclin F) reduced p130 levels in a dose-dependent manner (*Figure 3A*).

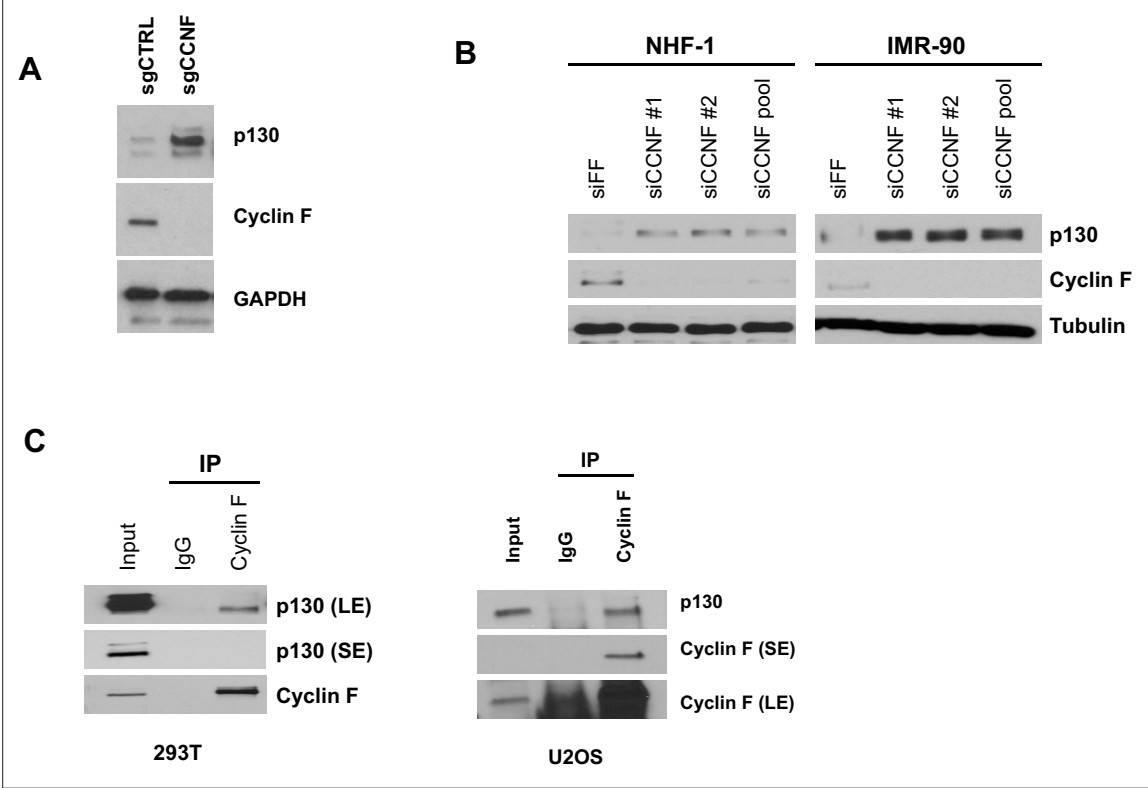

**Figure 2.** Cyclin F regulates and interacts with endogenous p130. (**A**) Asynchronously proliferating *CCNF* CRISPR/Cas9 knockouts (sgCCNF) and control (sgCtrl) HeLa cells were blotted for levels of the indicated proteins. Representative of n=3 experiments. (**B**) NHF-1 and IMR-90 cells were transfected with two different siRNAs targeting *CCNF* or a control siRNA targeting Firefly Luciferase (siFF). Whole-cell lysates were immunoblotted for the indicated proteins. Representative of n=3 experiments. (**C**) Endogenous cyclin F was immunoprecipitated from asynchronously proliferating HEK293T cells (left) or asynchronously proliferating U2OS cells (right). Indicated proteins were immunoblotted. SE=short exposure; LE=long exposure; representative of n=3 experiments.

The online version of this article includes the following figure supplement(s) for figure 2:

**Figure supplement 1.** Cyclin F knockout and depletion across cell lines.

Four human SCF E3 ligases are involved in marking substrates for degradation to promote entry and progression through S-phase: SCF^cyclin F, SCF^Skp2, SCF^βTRCP, and SCF^Fbxw7. We therefore transiently expressed HA-p130 alone or together with cyclin F, Skp2, Fbxw7, βTRCP1, or βTRCP2, and in both HEK293T and U2OS cells. p130 levels decreased only upon co-expression with cyclin F and not with any of the other F-box proteins (*Figure 3B* and *Figure 3—figure supplement 1A*).

In HEK293T cells, the reduction in HA-p130 caused by FLAG-cyclin F expression is reversed by addition of MG132 or MLN4924 (*Figure 3C*). Cyclin F recognizes its substrates through a hydrophobic patch motif in its cyclin domain (sequence MRYIL at amino acids M309-L313). Previous studies have shown that mutating the methionine and leucine residues to alanine (Cyclin F M309A L313A) can impair cyclin F binding to substrates (*Choudhury et al., 2016*; *D'Angiolella et al., 2012*; *D'Angiolella et al., 2010*). Notably, whereas cyclin F (WT) robustly downregulated p130 protein levels, cyclin F (M309A L313A) was unable to promote p130 degradation (*Figure 3D*). These data suggest that p130 is a cyclin F substrate.

## p130 is a cyclin F substrate

Cyclin F recognizes Cy Motifs in substrates, which corresponds to the amino acid sequence RxL/I (where x=any amino acid)(*Choudhury et al., 2016*; *D'Angiolella et al., 2012*; *D'Angiolella et al., 2010*). There are 13 putative Cy motif sequences in p130 that span the length of the protein. We therefore created six HA-tagged p130 truncations, while considering p130 domain structure (*Figure 4A*). We transiently expressed each HA-p130 truncation alone, or together with FLAG-cyclin F (WT) and included conditions with and without proteasome and neddylation inhibitors. p130 degradation

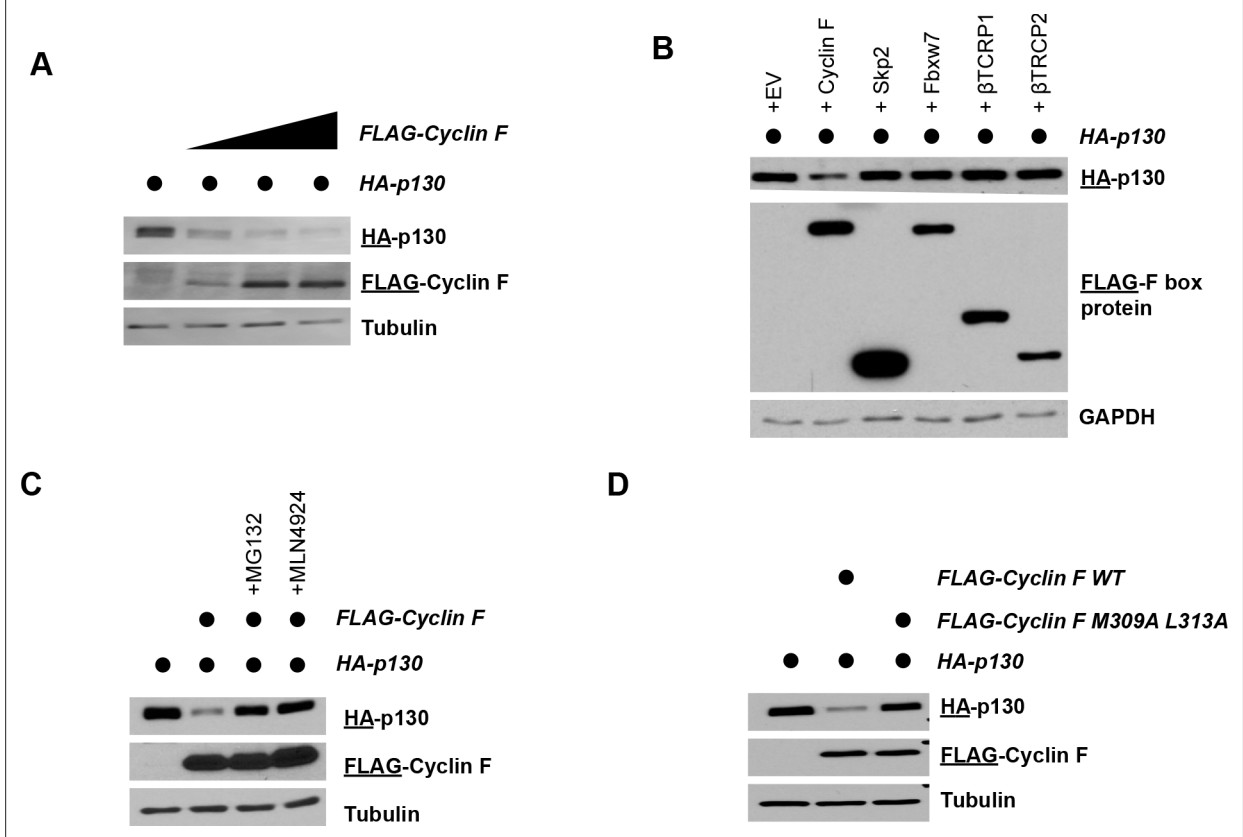

**Figure 3.** Cyclin F promotes p130 degradation. (**A**) HEK293T cells transiently expressing HA-p130 with an empty FLAG vector control (lane 1) or together with increasing amounts of FLAG-cyclin F (lanes 2–4). Cells were collected and analyzed by immunoblot 24 hr post-transfection. The antigen being immunoblotted for is represented by the underline, here and in all experiments below. Representative of n=3 experiments. (**B**) HEK293T cells transiently expressing HA-p130 with an empty FLAG vector control (lane 1) or together with FLAG-cyclin F (lanes 2–4). MG132 (proteasome inhibitor) or MLN4924 (neddylation inhibitor) were added for 6 hr prior to harvesting. Cells were collected and analyzed by immunoblot 24 hr post-transfection. Representative of n=3 experiments. (**C**) HEK293T cells transiently expressing HA-p130 with an empty FLAG vector control (lane 1) or together with FLAG-cyclin F WT (lane 2) or FLAG-cyclin F(M309A L313A) (lane 3), as indicated. Cells were collected and analyzed by immunoblot 24 hr post-transfection. Representative of n=3 experiments. (**D**) HEK293T cells transiently expressing HA-p130 with an empty FLAG vector control (lane 1) or together with the indicated FLAG-tagged F-box proteins (lanes 2–6). Cells were collected and analyzed by immunoblot 24 hr post-transfection. Representative of n=3 experiments.

The online version of this article includes the following figure supplement(s) for figure 3:

**Figure supplement 1.** P130 levels following Fbox protein expression in U2OS cells.

depended on the presence of a flexible spacer region, located between regions A and B in the p130 pocket domain (**Figure 4A**, **Figure 4—figure supplement 1A**). Within the spacer, there are two RxL/I motifs: R658-I660 and R680-L682. We mutated the first and last amino acid in each motif to alanine (AxA) in full-length HA-p130. Only the p130 (R680A L682A) mutant was resistant to cyclin F mediated degradation, demonstrating that these two amino acids alone are required for cyclin F to trigger p130 degradation (**Figure 4B**). That sequence and many surrounding residues are conserved in p130 proteins found in other species, including clawed frogs (*Xenopus tropicalis*), chickens (*Gallus gallus*), and mice (*Mus musculus*) (**Figure 4B**).

Since p130(R680A L682A; hereafter referred to as p130(AA)) cannot be degraded by cyclin F, we asked whether it still binds to cyclin F. GST-tagged p130 truncations were purified from *Escherichia coli* (amino acids 593–790; hereafter referred to as GST-p130), and mixed with total lysates from HEK293T cells expressing FLAG-cyclin F. In a GST-pulldown assay, FLAG-cyclin F bound to GST-p130(WT), but not to GST-p130(AA) or GST alone (**Figure 4C**, left panel). Conversely, FLAG-cyclin F, purified from HEK29T cells precipitated GST-p130(WT) but not GST-p130(AA) (**Figure 4C**, right panel). Consistent with its inability to trigger p130 degradation, cyclin F(M309A L313A) could not bind to GST-p130

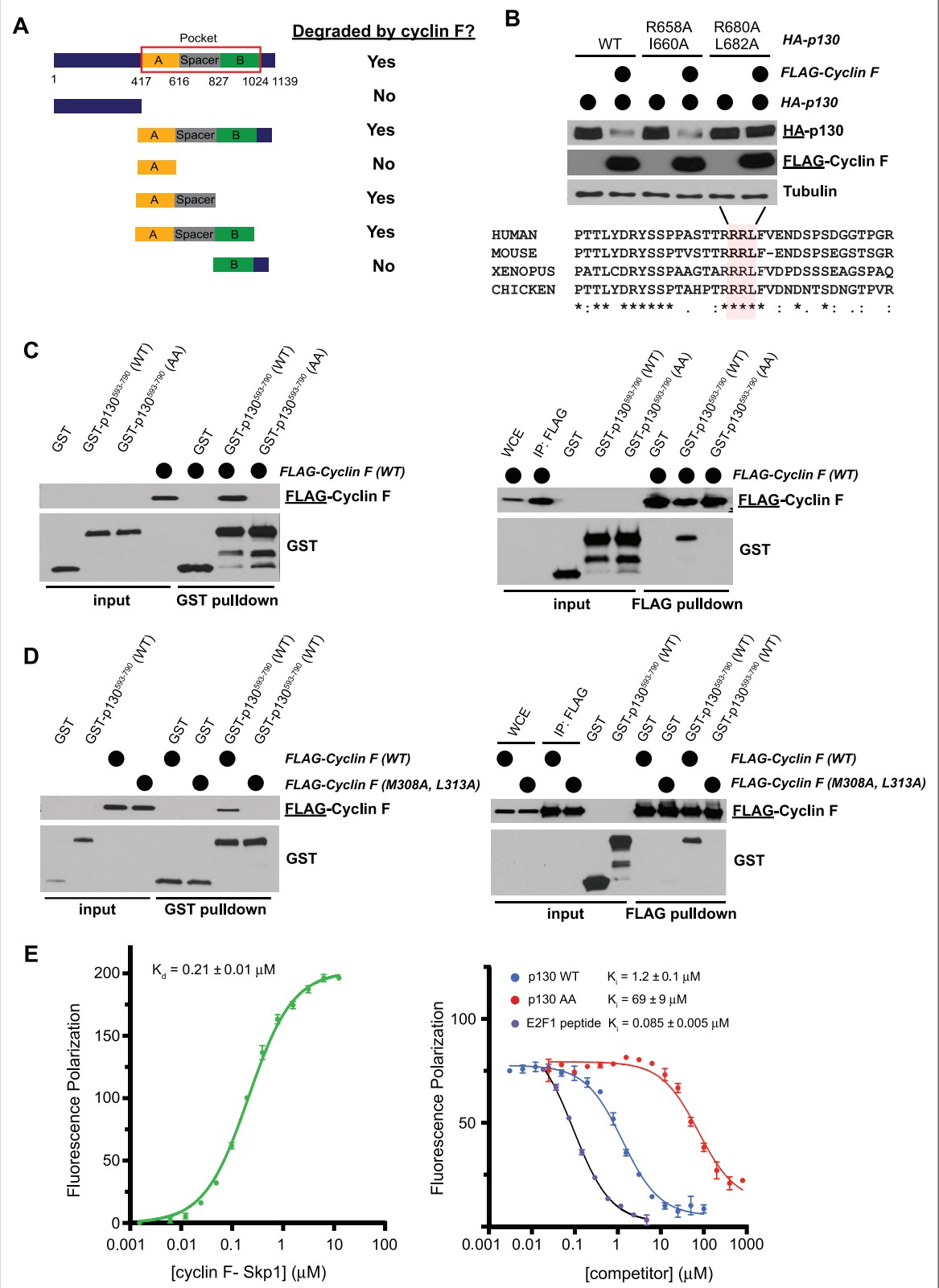

**Figure 4.** Cyclin F binds p130 directly and promotes p130 degradation through a conserved degron motif. (**A**) Graphical depiction of p130 domain structure. Indicated p130 truncation mutants were screened for their ability to be degraded following co-overexpression with cyclin F. The data supporting these conclusions are shown in *Figure 4—figure supplement 1*. (**B**) The first and last amino acids in the two potential cyclin F binding sites in the p130 spacer domain (R658-I660 and R680-L682) were mutated to alanine (AxA). HEK293T cells transiently expressed HA-p130 alone (WT or AxA

*Figure 4 continued on next page*

Figure 4 continued

mutants, as indicated) or together with FLAG-cyclin F WT. Cells were collected and analyzed by immunoblot 24 hr post-transfection. Representative of n=3 experiments (top) and amino acid sequence alignment for human, mouse, frog, and chicken p130 (bottom). (**C**) GST-p130$^{593-790}$ (WT) and GST-p130$^{593-790}$ (AA) were produced in *Escherichia coli* and purified. FLAG-cyclin F was transiently expressed in HEK293T cells. GST pulldowns (left) and FLAG pulldowns (right) were used to determine the interaction of p130 and cyclin F. Interaction was assessed by immunoblot after pulldown (representative of n=3 experiments). (**D**) GST-p130(WT) and FLAG-cyclin F(WT) or FLAG-cyclin F(M309A L313A) were expressed as described in (**C**). Interaction and binding were assessed as in (**C**) (representative of n=3 experiments). (**E**) Fluorescence polarization anisotropy assay to detect direct association of p130 with cyclin F. (Left) 10 nM TAMRA-p130$^{674-692}$ probe was titrated with increasing concentrations of purified GST-cyclin F$^{25-546}$-Skp1. (Right) The p130 probe bound with 0.5 µM GST-cyclin F$^{25-546}$-Skp1 was displaced with increasing concentrations of the indicated p130 protein construct or E2F1$^{84-99}$ peptide. Experiments were performed in triplicate, and the standard deviation is reported as the error.

The online version of this article includes the following figure supplement(s) for figure 4:

**Figure supplement 1.** Requirements for cyclin F binding and degradation of p130 in vivo and in vitro.

(*Figure 4D*). Thus, the p130 Cy motif at R680-L682 and the cyclin F hydrophobic patch (M309-L313) are required for the p130-cyclin F interaction and for p130 degradation.

To assess the direct binding and affinity of p130 for cyclin F, we developed a fluorescence polarization anisotropy assay using a peptide containing the relevant Cy motif in p130. We used a version of cyclin F, produced in Sf9 cells, which lacks the first ~25 amino acids and its C-terminal PEST domain to improve solubility (cyclin F$^{25-546}$). We titrated recombinant, purified cyclin F$^{25-546}$-Skp1 dimer into a solution of synthetic, TAMRA-labeled, p130 peptide (residues 674–692) and observed a $K_d$ of 0.21±0.01 µM (*Figure 4E*, left panel). Binding was efficiently competed away using purified GST-p130(WT) protein but not using GST-p130(AA), which had >50-fold higher $K_i$ in the competition assay (*Figure 4E*, right panel). We found that an unlabeled E2F1 peptide containing the relevant Cy motif (residues 84–99), which was previously shown to mediate cyclin F-E2F1 association (*Clijsters et al., 2019*), also competed with the p130 probe. We performed a similar competition assay using a TAMRA labeled E2F1$^{84-99}$ peptide as a probe, and we found that we could efficiently compete the E2F1 peptide with GST-p130(WT) but not GST-p130(AA) (*Figure 4—figure supplement 1B*). We conclude that p130 directly binds cyclin F-Skp1 at a site that overlaps with the E2F1 binding site and that the association is dependent on critical interactions made by R680 and L682 in p130.

To analyze p130 ubiquitination, we reconstituted the SCF$^{cyclin F}$ ubiquitination reaction in vitro using purified components. Cyclin F$^{25-546}$ was combined with neddylated Cul1-Roc1(Rbx1), Skp1, substrate, and ubiquitin. The human homolog of ariadne (ARIH1) is a RING-Between-RING (RBR) E3 ligase that has been shown to work with the E2 UBCH7 and other cullin ring E3 ligases, including the SCF, to ubiquitinate substrates (*Horn-Ghetko et al., 2021*; *Scott et al., 2016*). Therefore, ARIH1/UBCH7 and the chain-elongating E2 CDC34B were used as the ubiquitin transfer module.

SCF$^{Cyclin F}$ robustly ubiquitinated GST-p130(WT) (*Figure 5A*). This modification was dependent on the inclusion of NEDD8-Cul1-Roc1, Skp1-Cyclin F, and ARIH1/UBCH7, since the exclusion of any of these components completely abrogated ubiquitination (*Figure 5A*). Significantly, GST-p130(AA) was unable to be ubiquitinated by SCF$^{cyclin F}$.

Because cyclin F can ubiquitinate p130(WT), we determined the stability of p130(WT) and p130(AA) in the absence or presence of cyclin F. To determine p130 half-life, we expressed full-length HA-p130(WT) or HA-p130(AA) with or without FLAG-cyclin F in HEK293T or U2OS cells for 24 hr. Cells were treated with the protein synthesis inhibitor cycloheximide (CHX) and then analyzed for HA-p130 half-life. While p130(WT) half-life was significantly reduced by ectopic expression of cyclin F, p130(AA) remained stable for the length of the experiment in the presence or absence of cyclin F (*Figure 5B–C* and *Figure 5—figure supplement 1A*), indicating that protein levels of p130(AA) are resistant to cyclin F-mediated degradation.

We asked whether p130(AA) accumulates more than p130(WT) during a specific cell cycle phase. To evaluate p130 levels at different points during the cell cycle, we first engineered NHF-1 cells to express HA-tagged p130(WT) or p130(AA) in response to doxycycline treatment (*Figure 5—figure supplement 1B*). NHF-1 cells expressing p130 were synchronized with vehicle control, aphidicolin (S-phase synchronization), or nocodazole (Mitosis synchronization) for 16 hr. Interestingly, p130(WT) levels are decreased in nocodazole-synchronized cells compared to p130(AA) levels (*Figure 5D*). Consistently, nocodazole synchronization significantly decreased endogenous p130 protein levels in

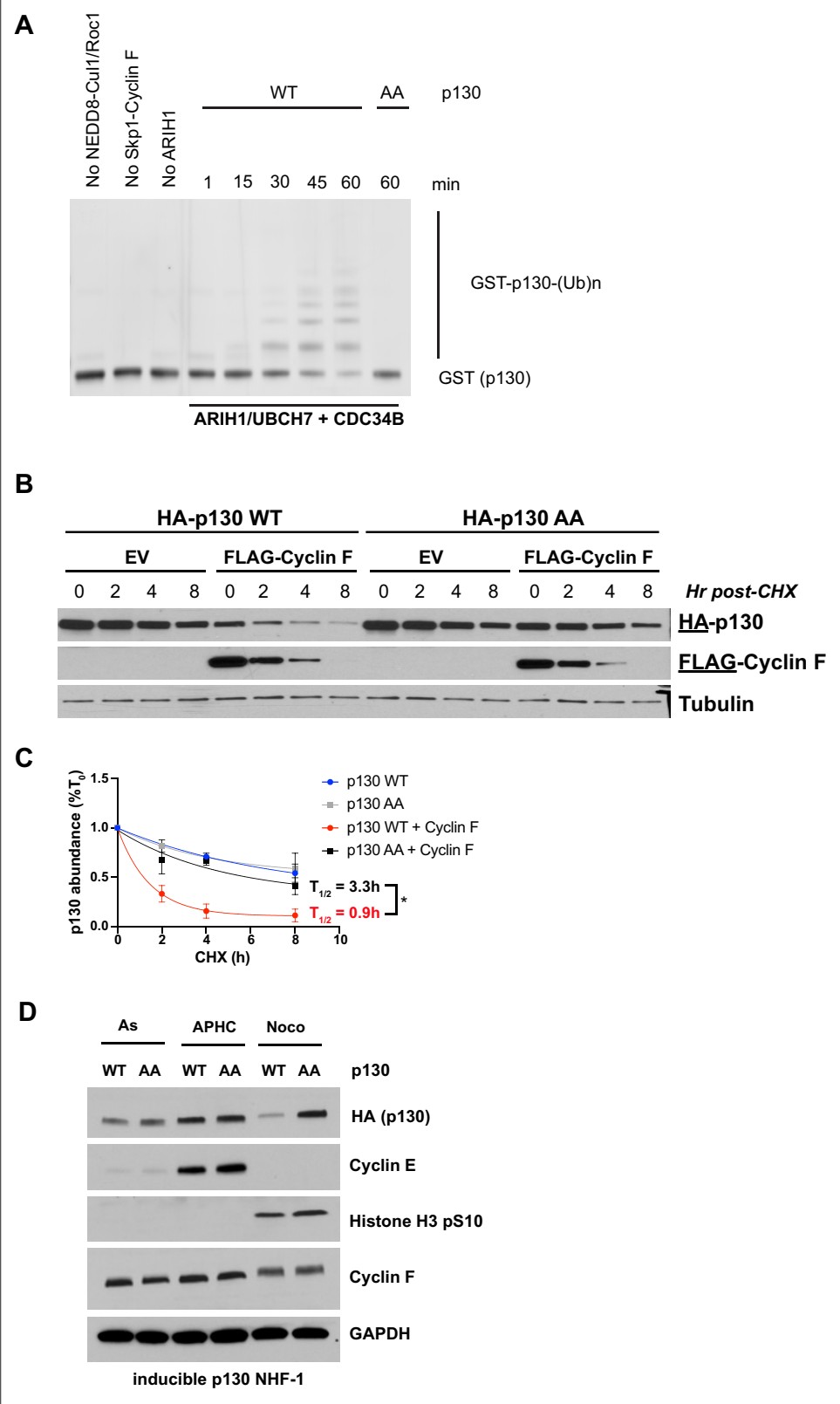

**Figure 5.** SCF[Cyclin F] regulates the ubiquitination and stability of p130. (**A**) Ubiquitination reactions were performed with SCF[cyclin F], ARIH1/UBCH7, and CDC34B. GST-tagged p130[593–790] WT and GST-p130[593–790] were used as substrates and detected by immunoblot against GST. Data are representative of n=3 experiments. (**B**) FLAG-cyclin F, HA-p130(WT), and/or HA-p130(AA) were transiently expressed in HEK293T cells for 24 hr. The protein synthesis

*Figure 5 continued on next page*

*Figure 5 continued*

inhibitor cycloheximide (CHX) was added and cells were collected at the indicated time points. Protein levels were determined by immunoblot. Representative of n=3 experiments. (**C**) Quantification of (**B**). *=p<0.05 (Student's t-test). Data are shown as mean ± SEM for n=3 experiments. (**D**) Inducible p130 NHF-1 cells were grown in media-containing 100 ng/ml doxycycline for 14 days to induce p130 expression, or in media-containing vehicle control. On indicated days, cells were pulsed with EdU for 30 min prior to harvest/fixation, DNA was stained with DAPI, and cells were analyzed by flow cytometry. Data for S- and G2/M phases are in ***Figure 6—figure supplement 1D***. Data represent mean ± SEM for n=3 replicates.

The online version of this article includes the following figure supplement(s) for figure 5:

**Figure supplement 1.** p130 stability in U2OS and NHF-1 cells.

**Figure supplement 2.** Both p130(WT) and p130(AA) can be phosphorylated and can associate with cyclin E, cyclin A, the DREAM complex, Large T antigen, and E7 protein.

---

NHF-1 cells (***Figure 5—figure supplement 1C***). Taken Together, these data indicate that p130(WT) levels decrease in nocodazole synchronized NHF-1 cells, whereas levels of p130(AA) remain stable.

The Cy motif that we mapped at amino acids 680–682 was previously implicated in binding cyclins A and E in SF9 cell extracts (***Lacy and Whyte, 1997***). To evaluate binding in human cells, we utilized the NHF-1 cells engineered to express p130(WT) or p130(AA). We immunoprecipitated (IP) HA and found that similar amounts of cyclin A and E co-purify with HA-p130(WT) and HA-p130(AA) (***Figure 5—figure supplement 2A***). Reciprocally, following IP of endogenous cyclin A, both HA-p130(WT) and HA-p130(AA) co-immunoprecipitated (***Figure 5—figure supplement 2B***). Further, we assessed whether mutation of the p130 Cy motif would impair p130 phosphorylation. HA-tagged p130(WT) and p130(AA) immunoprecipitated from NHF-1 cells are similarly phosphorylated on the CDK site at S672 (***Figure 5—figure supplement 2C***). Additionally, HA-p130(WT) and HA-p130(AA) migrate similarly in SDS-PAGE, their migration was similarly increased by post-lysis phosphatase treatment with calf intestinal phosphatase (CIP), and this could be prevented by the addition of phosphatase inhibitors (***Figure 5—figure supplement 2D***). These findings are consistent with previous reports mapping additional cyclin E and A binding sites to sequences in the p130 amino-terminus, and cyclin D binding to a carboxy-terminal helix (***Castaño et al., 1998***; ***Hansen et al., 2001***; ***Topacio et al., 2019***). Further, HA-tagged p130(WT) and p130(AA) can associate with the DREAM complex based on the co-IP of Lin54 following an HA pulldown (***Figure 5—figure supplement 2C***). Finally, we determined that the Cy motif mutations did not impact p130 binding to large T-antigen in HEK293T cells or the E7 oncoprotein in HeLa cells (***Figure 5—figure supplement 2E-F***).

## Loss of P130 regulation by cyclin F causes G0/G1 arrest and apoptosis

Interfering with RB phosphorylation, by expression of a mutant harboring substitutions at CDK phosphorylation sites, impairs cell cycle progression (***Fry et al., 2004***; ***Lukas et al., 1997***). To similarly isolate the phenotypic impact of cyclin F on p130, independent of other substrates, we utilized the NHF-1 cells engineered with a doxycycline inducible expression of p130(WT) or p130(AA) (***Figure 5—figure supplement 1A***). To determine whether p130(AA) accumulates more than p130(WT), we collected cells at various time points after induction. Although protein levels of p130(WT) and p130(AA) are initially equal, p130(AA) accumulated to higher levels compared to p130(WT) over time (***Figure 6A***), consistent with p130(AA) being resistant to degradation.

To assess proliferation, we induced expression of low levels of p130(WT) or p130(AA) and allowed cells to proliferate in culture for 14 days. Cells were split regularly, provided fresh media supplemented with doxycycline every 48 hr, and never reached confluency. Expression of non-degradable p130(AA) led to a profound decrease in proliferation/viability based on a PrestoBlue assay (***Figure 6—figure supplement 1A***). After 14 days, the PrestoBlue signal was decreased 5.8-fold (~83%) in p130(AA)-expressing cells relative to p130(WT) controls.

Next, we directly quantified proliferation by counting cells every 48 hr for 2 weeks after p130(WT) or p130(AA) induction. The p130(AA)-expressing cells proliferated much more slowly than the p130(WT)-expressing cells (***Figure 6B***). After 14 days, there were 7.7-fold (~87%) less cells in the p130(AA) population compared to p130(WT)-expressing controls. The average doubling time was 33 hr for p130(WT)-expressing cells and 48 hr for p130(AA)-expressing cells (***Figure 6C***). As a control, both cell lines were treated with vehicle alone. There was no difference in proliferation across a 2-week

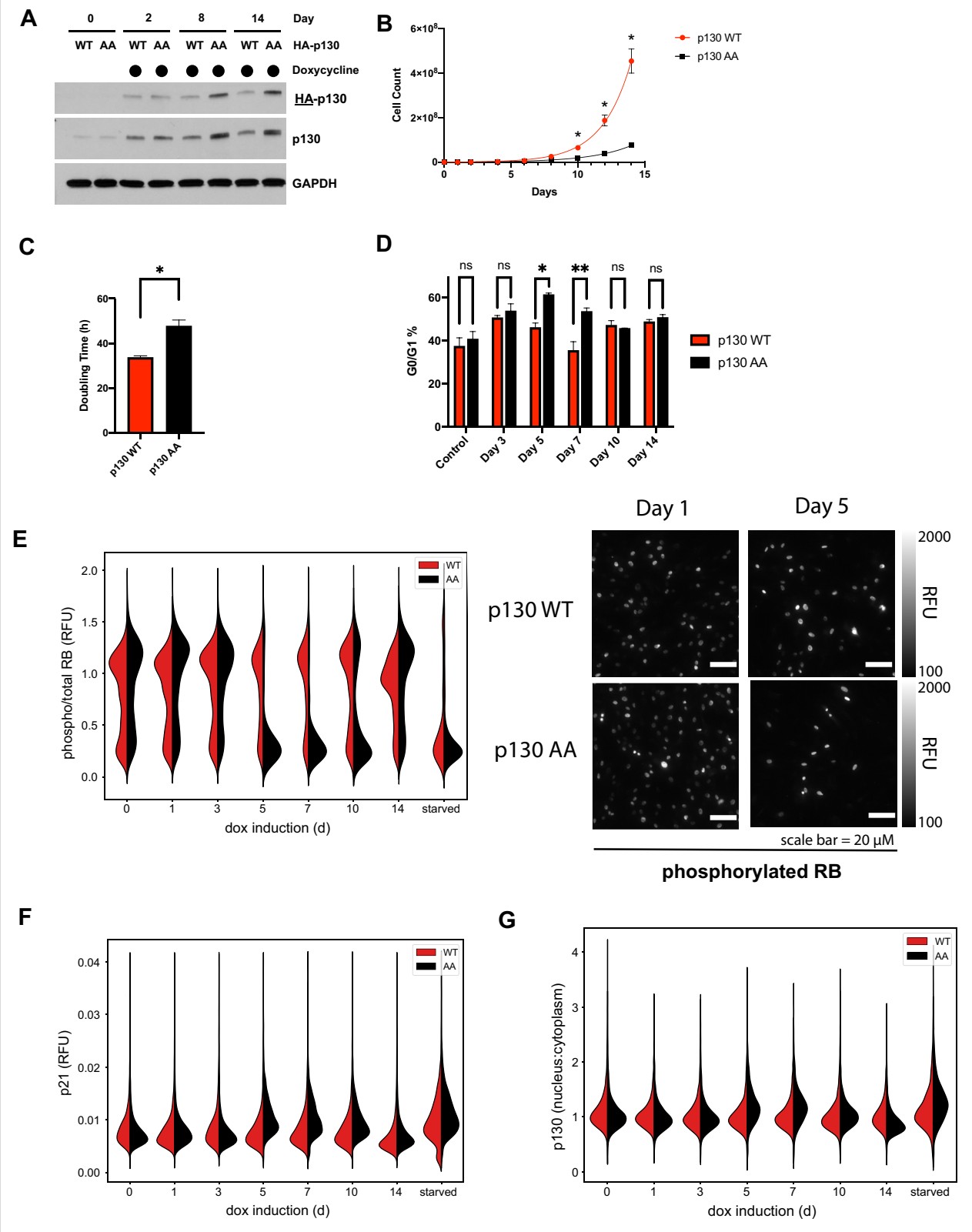

**Figure 6.** Cells expressing p130(AA) exhibit proliferation defects. (**A**) NHF-1 cells were engineered to express a TET-inducible HA-p130(WT) or HA-p130(AA) transgene. Doxycycline (100 ng/ml) was used to induce p130 expression, and water was used as a vehicle control. HA-p130 levels were assessed by immunoblot after cells were grown in doxycycline for up to 14 days. (**B**) Inducible p130 NHF-1 cells were grown in 100 ng/ml doxycycline-containing media for 14 days to induce p130 expression. Cells were counted on the indicated days. Data represent mean ± SEM for n=3 independent

*Figure 6 continued on next page*

*Figure 6 continued*

experiments. (**C**) Doubling time was calculated from counting experiment in (**B**). Error bars are SEM for n=3 independent experiments. Representative of n=2 independent experiments. (**D**) NHF-1 cells were grown in media supplemented with nocodazole for 16 hr, then immunoblotted for the indicated proteins. Representative of n=2 independent experiments. (**E–G**) Inducible p130 NHF-1 cells were grown in 100 ng/ml doxycycline for indicated times. Cells were fixed and analyzed by iterative immunofluorescent staining and imaging. Distributions of single-cell measurements are shown for nuclear phosphorylated versus total RB (**E**, left) and representative images of phosphorylated RB are shown for indicated days (**E**, right). Distributions of single-cell measurements are also shown for cytoplasmic p21 (**F**) and nuclear versus cytoplasmic p130 (**G**), as indicated.

The online version of this article includes the following figure supplement(s) for figure 6:

**Figure supplement 1.** TET-inducible p130(WT) and p130(AA) NHF-1 cells proliferate rapidly prior to induction of p130 expression.

**Figure supplement 2.** Cell cycle distribution and p130 staining from 4i experiments.

experiment (*Figure 6—figure supplement 1B-C*). Expectedly, ectopic expression of p130(WT) slowed the doubling time compared to cells treated with vehicle, consistent with previous studies which report p130 overexpression slows cell growth (*Lacy and Whyte, 1997*). However, p130(AA) had a much more severe impact on proliferation/survival.

We investigated whether the slow growth observed following p130(AA)-induction resulted from cell cycle arrest. We performed flow cytometry to analyze cell cycle phase distribution using EdU incorporation and DNA staining. NHF-1 were pulsed with EdU for 30 min prior to fixation at several time points after p130 induction. Total DNA was stained with DAPI, and cells were analyzed by flow cytometry. The percent of p130(AA)-expressing cells in G0/G1 significantly increased at days 5 and 7 compared to p130(WT)-expressing cells, while the percentage of S-phase cells was significantly decreased (*Figure 6D* and *Figure 6—figure supplement 1D*). Surprisingly, beyond day 10, there was no difference in G0/G1% between p130(AA) and p130(WT)-expressing cells, suggesting that the cell population may adapt to the increased levels of p130, which remain elevated throughout the entire 2-week experiment (*Figure 6D*).

We used iterative indirect immunofluorescence imaging (4i) to determine the levels of cell cycle markers in p130(WT)- and p130(AA)-expressing cells. 4i is accomplished by repeated antibody staining, followed by imaging, stripping, and re-staining, that collectively produce a set of targeted protein measurements in single cells. The resulting images are 'stacked' to quantify the expression of multiple proteins and antigens across an identical set of individual cells (*Gut et al., 2018*). Automated and semi-automated methods are used to identify cell boundaries and extract quantitative protein-level information for each cell (*Gut et al., 2018*).

To determine whether cells were arresting in G1, we stained for total and phospho-RB. RB is phosphorylated in proliferating cells, while unphosphorylated RB is a hallmark of G1/G0. RB phosphorylation was decreased in p130(AA)-expressing cells at days 5, 7, and 10 compared to p130(WT)-expressing cells or vehicle-treated controls (*Figure 6E*). We also found that p21 was increased in p130(AA)-expressing cells at days 5, 7, and 10, consistent with a G0/G1 arrest (*Figure 6F*). To assess cell cycle distribution in each population, we quantified total DNA content. DNA content analysis revealed that a larger percentage of p130(AA)-expressing cells had <4C DNA content compared to p130(WT)-expressing cells at days 5, 7, and 10, consistent with cells accumulating in G0/G1-phase (*Figure 6—figure supplement 2A*). Finally, we assessed the subcellular localization of p130 at each time point by quantifying the ratio of nuclear:cytoplasmic intensity from immunofluorescence images. We found p130(AA) was more localized to the nucleus at days 5 and 7 compared to p130(WT) (*Figure 6G* and *Figure 6—figure supplement 2B*), consistent with p130(AA) mediating cell cycle arrest through cell cycle gene repression.

As a component of the DREAM complex, p130 functions to prevent the transcription of myriad early (S-phase) and late (G2/M) cell cycle genes (*Engeland, 2018*). Because we observe an increase in G0/G1% for populations of cells expressing p130(AA), we sought to determine the expression levels of several DREAM target genes including: *CCNF, CCNE1, CDC6, DHFR, E2F1, and CDT1*. There was a significant reduction in expression of all of these in p130(AA)-expressing cells compared to those expressing p130(WT) (*Figure 7A*). Since both p130(WT)-expressing cells and p130(AA)-expressing cells grew more slowly than cells not expressing a p130 transgene, we also performed RT-qPCR on control cells; the expression of *CCNF, CCNE1, CDC6, DHFR, E2F1, and CDT1* was lower in both p130(WT)- and p130 (AA)-expressing cells compared to controls (*Figure 7—figure supplement 1A*). Thus, ectopic expression of p130 reduces transcription of DREAM target genes, which is further

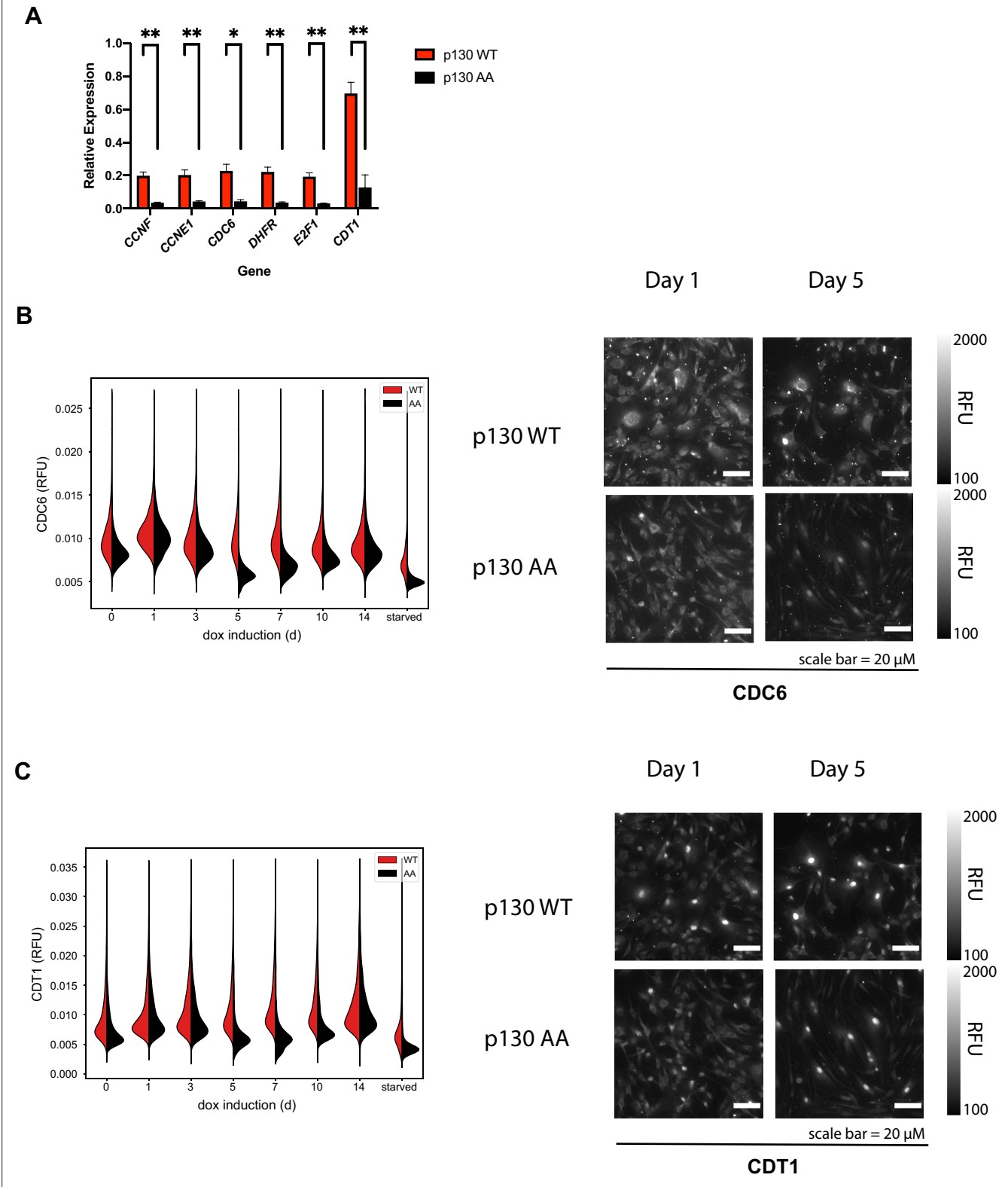

**Figure 7.** DREAM targets are downregulated in cells expressing p130(AA). (**A**) Inducible p130(WT) and p130(AA) NHF-1 cells were grown for 8 days in media-containing 100 ng/ml doxycycline to induce p130 expression. RNA was extracted for rt-qPCR analysis. Gene expression is relative to GAPDH expression and normalized to the vehicle control. Data are mean of n=3 experiments, and error bars are SEM. (**B, C**) Inducible p130 NHF-1 cells were grown in 100 ng/ml doxycycline for indicated times, and protein levels of CDC6 (**B**) and Cdt1 (**C**) were analyzed by iterative immunofluorescent staining

*Figure 7 continued on next page*

*Figure 7 continued*

and fixed cell imaging. Distributions of single-cell measurements of Cytoplasmic CDC6 (**B**) and nuclear CDT1 (**C**) were plotted (left) and representative images are shown for the indicated days (right).

The online version of this article includes the following figure supplement(s) for figure 7:

**Figure supplement 1.** RT-pPCR for DREAM target genes in NHF-1 cells expressing control versus p130(WT) versus p130(A).

**Figure supplement 2.** Apoptosis and DNA damage markers in cells expressing p130(WT) or p130(AA).

decreased by the expression of p130(AA). To confirm that the reduction in gene expression of DREAM targets led to decreased protein expression, we used 4i to quantify two DREAM targets by immuno-fluorescence: CDC6 and Cdt1. Consistent with DREAM activation, as well as observed G0/G1 arrest, protein levels of both CDC6 and Cdt1 were decreased from days 5 to 10 in p130(AA)-expressing cells compared to p130(WT)-expressing cells (*Figure 7B–C*).

Because not all cells arrest in G0/G1, we asked whether another factor may also be contributing to the reduced proliferation of the p130(AA)-expressing population. High p130 levels have previously been linked to apoptosis (*Pentimalli et al., 2018*; *Ventura et al., 2018*), whereas low p130 levels have been shown to protect against apoptosis (*Bellan et al., 2002*). We therefore examined the apoptotic markers cleaved PARP and cleaved caspase 3. As a positive control, cell lines were treated with 100 nM staurosporine to induce apoptosis. We observe cleaved PARP in the p130(AA)-expressing population at days 7, 10, and 14, and cleaved caspase 3 at days 7 and 10, indicating that cells in these populations are undergoing apoptosis. We did not observe cleaved PARP or cleaved caspase in any of the populations for p130(WT)-expressing cells (*Figure 7—figure supplement 2A*). Additionally, we performed flow cytometry for Annexin V and propidium iodide incorporation in unfixed cells to assess apoptosis/necrosis. We observed a higher percentage of apoptotic/necrotic cells in the p130(AA)-expressing population compared to the p130(WT)-expressing population at day 8 (*Figure 7—figure supplement 2B*). Since DNA damage can cause apoptosis, and because high p21 levels can be an indicator of DNA damage, we used 4i to determine the expression of proteins in the DNA damage response pathway, including 53BP1, phospho-Chk1, and phospho-H2A.X. We found that in the p130(AA)-expressing cells at days 5, 7, and 10, that levels of 53BP1, phospho-Chk1, and phospho-H2A.X were down-regulated (*Figure 7—figure supplement 2C-E*). A decrease in DNA damage markers is consistent with fewer cells being in S-phase and suggests that DNA damage is not the cause of apoptosis. We also examined senescence but saw no apparent increase in beta-galactosidase staining in p130(AA)-expressing cells.

## Discussion

We demonstrate that the RB-like protein p130 is a substrate of the E3 ubiquitin ligase SCF^{Cyclin F}. Cyclin F loss allows p130 to accumulate, whereas its overexpression promotes p130 degradation. Cyclin F and p130 bind directly, and SCF^{cyclin F} can ubiquitinate p130 in a fully reconstituted in vitro system. We mapped a critical cyclin F binding motif in p130, and when that sequence is mutated, it prevents p130 ubiquitination and degradation. This mutant version of p130 accumulates in normal human fibroblast cells, causing slowed proliferation, an accumulation in G0/G1-phase, and apoptosis.

Taken together, our results suggest that cyclin F contributes to inactivation of p130 and is therefore a regulator of the CDK-RB network. This finding is consistent with our observation that *CCNF* KO is correlated with *CDK4* and *CCND1/Cyclin D1* KO (*Figure 1—figure supplement 3*). Interestingly, the DepMap has tested hundreds of cell lines for their sensitivity to over 1000 oncology and non-oncology drugs. We queried these data for gene-drug similarities related to the selective CDK4/6 inhibitors Palbociclib and Ribociclib, which are used to treat metastatic, hormone receptor positive breast cancer (*Agostinetto et al., 2021*; *Kay et al., 2021*). Comparing Palbociclib to CRISPR knockout screens, revealed *CCND1/Cyclin D1 and CDK4* as the most highly correlated genes, validating this approach. Notably, *CCNF* ranked ninth among over 17,000 genes queried (*Figure 8A*). For Ribociclib, *CCND1* is the most highly correlated, *CDK4* is ranked 5th, and *CCNF* was 125th, falling within the top 1% of most highly correlated genes (*Figure 8B*). Conversely, we determined which drugs are highly correlated with *CDK4, CCND1, and CCNF* KO. Palbociclib and Ribociclib are among the top four most highly correlated drugs with *CDK4* and *CCND1* (*Figure 8C–D*). Remarkably, when *CCNF/Cyclin*

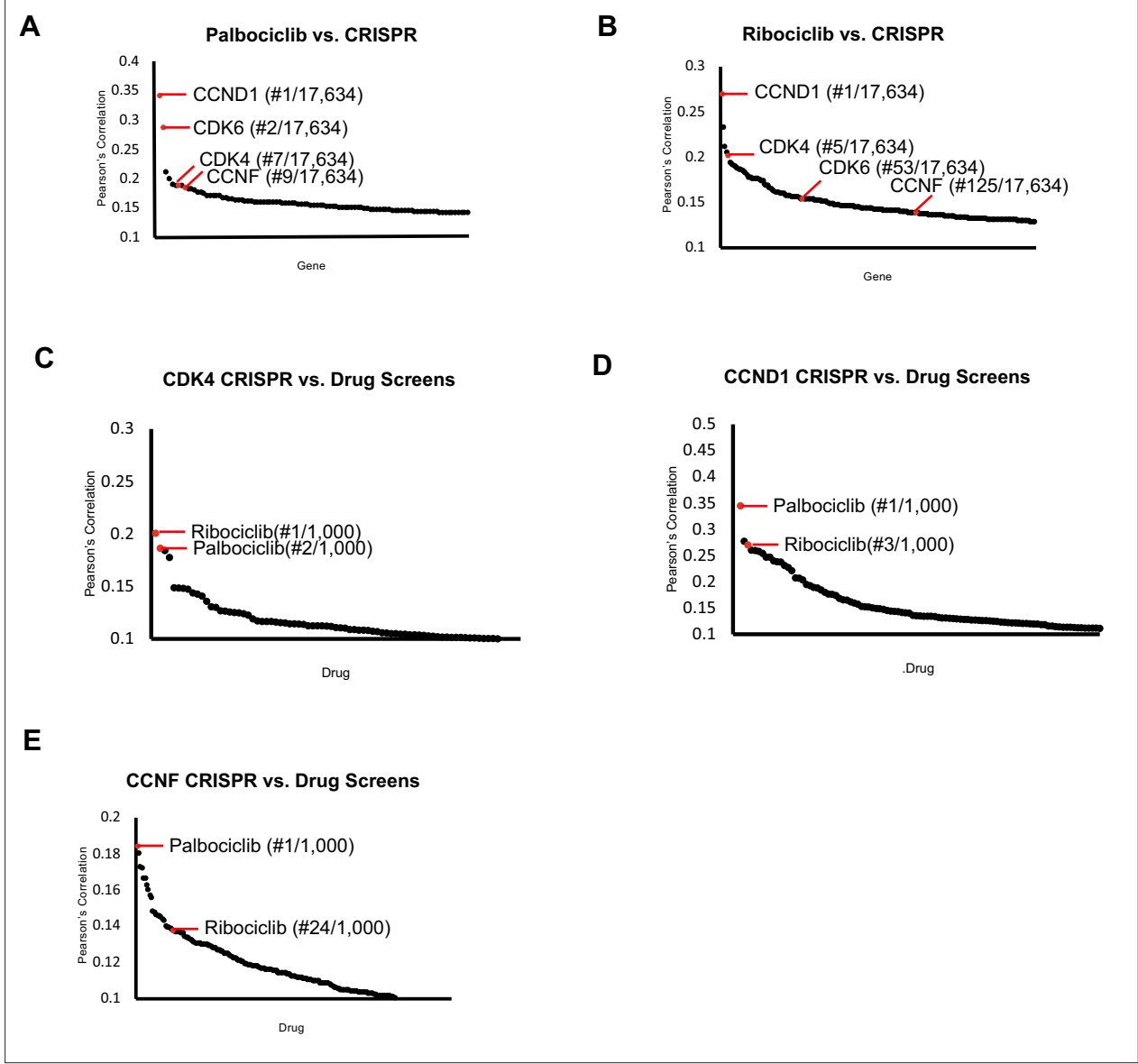

**Figure 8.** CDK4/6 inhibitors and CCNF knockout correlate highly. Project Achilles Dependency Map data sets were analyzed to determine: (**A**) Pearson's correlation coefficients for gene knockout correlated with Palbociclib treatment. (**B**) Pearson's correlation coefficients for gene knockout correlated with Ribociclib treatment. (**C**) Pearson's correlation coefficients for the correlation between *CDK4* knockout and 1000 drug treatments. (**D**) Pearson's correlation coefficients for the correlation between *CCND1* knockout and 1000 drug treatments. (**E**) Pearson's correlation coefficients for the correlation between *CCNF* knockout and 1000 drug treatments. For (**A–E**), only the top scoring, most statistically significant associations are shown.

*F* knockout is compared to all drugs tested, Palbociclib is ranked 1st, and Ribociclib is ranked 24th (***Figure 8E***). Thus, *CCNF* KO is highly correlated with both genetic and pharmacologic inactivation of the CDK-RB network across hundreds of cell lines.

## RB-family protein dynamics throughout the cell cycle

The activity of RB and the RB-like proteins, p107 and p130, oscillates during the cell cycle and quiescence. Cyclin-CDK complexes phosphorylate RB, p107, and p130, which prevents their binding to activator E2Fs, allowing for the transcription of cell cycle target genes. In addition, RB-like proteins are also controlled at the level of their abundance. As cells exit quiescence, p130 protein is degraded by the ubiquitin-proteasome system. Then, when cells transition from proliferation to quiescence, p130 protein levels significantly increase (***Smith et al., 1996***). The role of phosphorylation in controlling RB and the RB-like proteins has been well-established (***Canhoto et al., 2000***; ***Farkas et al., 2002***;

*Hansen et al., 2001*); however, much less is known about how changes in the levels of RB-like proteins might contribute to changes in pathway activity or their ability to restrain E2F transcription factors. Notably, as cells grow bigger, RB and its yeast ortholog Whi5, are diluted, contributing to cell cycle progression (*Schmoller et al., 2015*; *Zatulovskiy et al., 2020*). Our studies suggest that the targeted degradation of p130 by cyclin F, which rapidly reduces its concentration, also plays a significant role in promoting cell cycle progression.

Despite a concrete understanding of the role of phosphorylation in regulating RB-like proteins, less is known about the role of ubiquitin ligases. The phosphatase subunit NRBE3 was previously linked to RB ubiquitination (*Wang et al., 2015*). A nucleolar protein, U3 protein 14a (hUTP14a), was also suggested to be a novel type of E3 ubiquitin ligase, capable of promoting both RB and p53 degradation and cancer cell proliferation (*Liu et al., 2018*). However, neither NRBE3 nor hUTP14a are ubiquitin ligases. Finally, RB degradation by an unknown, Cul2-based E3 was shown in cells expressing HPV E7 oncoprotein (*White et al., 2012*). Previous studies have reported that SCF[Skp2] recognizes p130 and promotes its ubiquitination (*Bhattacharya et al., 2003*; *Tedesco et al., 2002*). However, p130 still cycles in Skp2 knockout cells, suggesting additional E3 ligase(s) may be involved in p130 regulation. Surprisingly, for unknown reasons, Skp2 overexpression had no effect on p130 levels in our experiments.

## Cyclin F as a regulator of the cell cycle gene transcription

Cyclin F has previously been linked to cell cycle gene transcription via its regulation of several proteins, including B-MYB, SLBP, E2F1, E2F2, E2F3a, E2F7, and E2F8 (*Emanuele et al., 2020*). In response to DNA damage, cyclin F is able to bind to B-MYB, and rather than ubiquitinate it, inhibits the ability of B-MYB to promote the expression of mitotic genes (*Klein et al., 2015*). In G2, cyclin F ubiquitinates the stem-loop binding protein (SLBP), which is required during S-phase to mediate histone biogenesis (*Dankert et al., 2016*). Also, during G2, cyclin F catalyzes the ubiquitination of the activator E2Fs: E2F1, E2F2, and E2F3a (*Burdova et al., 2019*; *Clijsters et al., 2019*). The ubiquitination of activator E2Fs allows for their degradation, and expression of mutant E2Fs that can no longer bind to cyclin F results in increased expression of E2F target genes. Further, during G2-phase, cyclin F was shown to regulate the atypical repressor E2Fs, E2F7, and E2F8, promoting their degradation (*Wasserman et al., 2020*; *Yuan et al., 2019*). Each of these substrates implicates cyclin F in indirectly controlling cell cycle gene expression, and our study uncovers a new mechanism by which cyclin F regulates transcriptional control at the start of the cell cycle, by ubiquitinating p130.

Adding to the complexity of the role of cyclin F in cell cycle transcription, not only does cyclin F regulate p130 degradation, but p130 regulates the expression of cyclin F. Thus, cyclin F and p130 appear to exist in a double negative feedback loop, where each can downregulate the other. Interestingly, in response to DNA damage, cyclin F is downregulated by degradation (*D'Angiolella et al., 2012*), and p130 has been suggested to become activated to repress cell cycle gene expression. Thus, the degradation of cyclin F in response to genotoxic stress could preserve cell cycle arrest by allowing for the accumulation of p130, while also contributing to increased nucleotide pools through the accumulation of another cyclin F substrate, RRM2 (*D'Angiolella et al., 2012*).

## Effect of P130 overexpression on proliferation

We have shown that expressing the mutant p130(AA), but not p130(WT), causes growth arrest and apoptosis. These results are consistent with clinical observations of outcomes of p130 expression levels in cancerous tissues. In lung cancers, p130 expression negatively correlates with histological grading and metastasis ( *Baldi and Esposito, 1997*). For endometrial cancer patients who receive surgery, low p130 levels are significantly associated with increased recurrence and death (*Susini et al., 1998*). In vulvar carcinomas, loss of p130 and p27 has been shown to contribute to carcinogenesis (*Zamparelli et al., 2001*). In oral squamous cell carcinoma, p130-negative cases have worse prognoses than p130-positive cases (*Tanaka et al., 2001*). And, finally, in thyroid neoplasms, reduced p130 expression has been linked to the aggressive characteristics of anaplastic carcinoma, while high p130 expression in micropapillary carcinoma has been linked to the smaller size of those tumors (*Ito et al., 2003*). Thus, cancers with low p130 expression or p130 loss, seem to grow more quickly and aggressively, whereas cancers with high p130 expression are correlated with better prognoses. Our findings fit within the context of these studies; as p130(AA) accumulates, the cells are less proliferative and some die by

apoptosis. Since p130 mRNA levels are not regulated during the cell cycle, it is interesting to speculate that enhancing p130 degradation in some malignancies could contribute to defects in the regulation of cell cycle gene expression and hyper-proliferation. Likewise, blocking p130 degradation could phenocopy CDK4/6 inactivation and could potentially be therapeutically advantageous, and would be consistent with our analysis showing that loss-of-function in cyclin D, CDK4, and cyclin F are highly correlated across nearly 800 cancer cell lines.

Cyclin F is a strongly cell cycle-regulated gene. Cyclin F mRNA levels peak in G2/M, and no studies have identified *CCNF* as an E2F target at G1/S. It is therefore unclear how its protein levels increase early in the cell cycle, in lockstep with Skp2, a bona fide E2F target gene. We previously showed that AKT can phosphorylate and stabilize cyclin F, and that this could contribute to its accumulation near G1/S (*Choudhury et al., 2017*). Thus, the activation of cyclin F by AKT could potentially lead to the degradation of p130. Given the recurrent activation of the PI3K-AKT pathway in many cancers, it will be interesting in the future to determine if cyclin F phosphorylation promotes p130 degradation. If true, this would suggest that hyper-activated PI3K, either through mutation of PIK3CA or loss of PTEN, functions through cyclin F to render p130 inactive via the ubiquitin pathway.

## Materials and methods
### Acquisition and analysis of DEPMAP data
Gene co-dependencies were determined from the Achilles data set from depmap.org (Achilles_gene_effect.csv, downloaded 7/19/19). The Achilles data set contains dependency scores from genome-scale essentiality screens scores of 789 cell lines. As a measure of co-dependency, the Pearson's correlation coefficient of essentiality scores was computed for all gene pairs. GO analysis for the top 100 genes co-dependent with CCNF was performed using MetaScape. Potential cyclin F substrates were identified by proteins encoded by genes that have a Pearson's correlation coefficient of >0.15 when compared to CCNF and are classified by the GO term: cell division (0051301).

### Cell lines
Cell lines used include: HEK293T (ATCC), U2OS (ATCC), HeLa (ATCC), HeLa sgCTRL, and HeLa sgCCNF (*Choudhury et al., 2016*), MCF-7 (ATCC), T47D (ATCC), NHF-1 William Kaufman Lab (UNC; retired), IMR-90 (Yue Xiong Lab, UNC; retired), and T98G (Tissue Culture Facility, UNC). Additional cell line details are listed in the table in Appendix 1. Cell line authenticity was verified by SPR. All cell lines were checked for mycoplasma contamination periodically throughout the experimental process.

### Cell culture
All cell lines were cultured in Dulbecco's modified Eagle's medium (DMEM; GIBCO) supplemented with 10% fetal bovine serum (FBS; VWR) and 1% Pen/Strep (GIBCO). Cells were incubated at 37°C, 5% $CO_2$.

DNA transfection experiments were performed in HEK293T cells for 24 hr using either Lipofectamine 2000 (Thermo Fisher Scientific) or PolyJet (SignaGen) transfection reagents according to the manufacturer's protocol. RNA transfection experiments were performed in NHF-1 and IMR-90 cells for 48 hr using RNAiMAX (Thermo Fisher Scientific) transfection reagent according to the manufacturer's protocol. Plasmid and siRNA information is the table in Appendix 1.

To produce lentivirus, HEK293T cells were transfected with pLV[Exp]-CMV> Tet3G/Hygro, pLV[TetOn]-Neo-TRE3G > HA/{p130 WT}, or pLV[TetOn]-Neo-TRE3G > HA/{p130 AA}, and the lentivirus packaging plasmids VSV-G, Gag-pol, Tat, and Rev. Media-containing virus particles were collected after 48 hr, filtered with a 0.45-µM filter, and frozen at –80°C. NHF-1 cells were transduced with a 1:1 ratio of fresh media plus lentivirus-containing media. NHF-1 cells were first transduced with the Tet3G plasmid and selected in media-containing 50 µg/ml hygromycin. Then, NHF-1 cells stably expressing Tet3G were infected with either p130 WT or p130 AA and selected in media-containing 375 µg/ml G418.

Cloning and Directed Mutagenesis pDEST-HA3-p130, pDEST-HA3-p130$^{1–417}$, pDEST-HA3-p130$^{418–1139}$, pDEST-HA3-p130$^{418–616}$, pDEST-HA3-p130$^{418–827}$, pDEST-HA3-p130$^{418–1024}$, pDEST-HA3-p130$^{828–1139}$, and pINDUCER20 CCNF were produced using Gateway Cloning Technology. pLV[Exp]-CMV> Tet3G/Hygro, pLV[TetOn]-Neo-TRE3G > HA/{p130 WT}, or pLV[TetOn]-Neo-TRE3G > HA/{p130 AA} were

all ordered from VectorBuilder. pGEX-GST-p130$^{593–790}$ was a gift from Peter Whyte (McMaster University) and pINDUCER20 was a gift from Stephen Elledge (Addgene #44012). pDEST-HA3-p130 R658A I660A, pDEST-HA3-p130 R680A L682A, and pGEX-GST-p130$^{593–790}$ R680A L682A were generated by site-directed mutagenesis using the Q5 Site-Directed Mutagenesis Kit (NEB) according to the manufacturer's instructions. For recombinant expression, Cyclin F$^{25–546}$ was subcloned into a pFastBac vector with an N-terminal GST tag and TEV protease site. The Skp1 gene (a gift from Dr. Bing Hao) was subcloned into a PGEX-4T-1 vector previously engineered to contain a TEV protease site. Sequences for all primers are included in tables provided in Appendix 1.

## Cell lysis and immunoblotting

Cells were lysed on ice for 10 min in NETN (20 mM Tris pH 8.0, 100 mM NaCl, 0.5 mM EDTA, 0.5% NP40) supplemented with 10 μg/ml aprotinin, 10 μg/ml leupeptin, 10 μg/ml pepstatin A, 1 mM sodium orthovanadate, and 1 mM AEBSF(4-[two aminoethyl] benzenesulfonyl fluoride). Following incubation, cells were spun at 20,000×$g$ in a benchtop microcentrifuge at 4°C for 10 min. Protein concentration was determined by Bradford assay (Bio-Rad) and samples were prepared by boiling in Laemmli buffer (LB). Protein was separated by electrophoresis on either homemade or TGX (Bio-Rad) stain-free gels and then was transferred to nitrocellulose membranes. Blocking was performed in 5% non-fat dry milk (Blotting Grade blocker; Bio-Rad) diluted in TBS-T (137 mM NaCl, 2.7 mM KCl, 25 mM Tris pH 7.4, 1 % Tween-20). All primary antibody incubations were carried out overnight, rocking, at 4°C, and all HRP-conjugated antibody incubations were carried out for 1 hr, rocking, at room temperature. TBS-T was used for all wash steps. Protein abundance was visualized by chemiluminescence using Pierce ECL (Thermo Fisher Scientific). A detailed list of antibodies and dilutions is provided in table in Appendix 1.

## Immunoprecipitation

### Endogenous protein pulldowns:

Asynchronous HEK293T, U2OS, HeLa, or inducible p130 NHF-1 cells were lysed and total protein was quantified as described above. 10% of the protein sample was retained as the input, and the remaining protein was divided in half to be incubated with either rabbit IgG control or Cyclin F antibodies for IP. Cell lysates were incubated for 6 hr, rotating, at 4°C with 1 μg antibody per mg protein. During the incubation, Pierce Protein A/G agarose beads (Thermo Fisher Scientific) were washed 3× for 20 min in NETN, then resuspended at a 50% slurry in NETN. 40 μl slurry per 1 mg protein was added to the protein/antibody IP mix and incubated for 45 min, rotating, at 4°C. Beads were then washed 3× for 5 min in NETN, and finally re-suspended in 2× LB and boiled. Co-IP was assessed by immunoblotting as described.

### Exogenous pulldowns:

Asynchronously proliferating inducible p130 NHF-1 cells were treated with 100 ng/ml doxycycline for 24 hr. Cells were lysed and total protein was quantified as above. 10% of the protein sample was retained as the input, and the remaining protein was divided in half to be incubated with either rabbit IgG control or rabbit anti-HA antibodies for IP. IP and washes were carried out as described above. Alternatively, inducible p130 NHF-1 cells were treated with 100 ng/ml doxycycline or water as a vehicle control for 24 hr. Cells were lysed and total protein was quantified as described above. 50 μl of EZView Red anti-HA affinity gel slurry was used to isolate HA-tagged and co-immunoprecipitating proteins. Washes were performed as described above. Co-IP was assessed by immunoblotting.

## Protein purification

In order to purify GST and GST-p130 for the in vitro binding assay, BL21 *E. coli* were transformed with GST, GST-p130$^{WT}$ or GST-p130$^{AA}$ and a 0.5-L culture was grown to an O.D. of 0.6 then induced with 1 mM IPTG (Isopropyl β-D-1-thiogalactopyranoside) for 4 hr. Cells were harvested by centrifugation and incubated for 20 min in NTEN *E. coli* Lysis buffer (100 mM NaCl$_2$, 20 mM Tris pH 8.0, 1 mM DTT, 1 mM EDTA, and 1% NP40) supplemented with 10 μg/ml aprotinin, 10 μg/ml leupeptin, 10 μg/ml pepstatin A, and 10 mg/ml lysozyme. Then, cells were lysed by sonication at 50% power for 3× for 30 s pulses, with 1 min on ice between pulses. Lysates were then spun at 30,000×$g$ for 30 min at 4°C to clarify. During the clarification step, glutathione agarose resin (GoldBio) was equilibrated 3× for 10 min in NTEN *E. coli* Lysis buffer. 250 μl of equilibrated glutathione agarose resin was then added to

the clarified lysate and incubated for 2 hr, rotating, at 4°C. Beads were washed 3× for 5 min in NTEN *E. coli* Lysis buffer. Bound GST-p130 was eluted by three sequential elution steps: 2× for 30 min and one overnight incubation, rotating, at 4°C in elution buffer (50 mM Tris pH 8.0, 5 mM reduced glutathione). Elution fractions were pooled, and buffer exchange into 50 mM Tris was performed by 5× buffer exchange in 30,000 MWCO spin columns.

For in vitro ubiquitination assays, UBA1, CDC34B, UBCH7, ARIH1, Cul1-Roc1, and ubiquitin were expressed and purified as described previously (*Kamadurai et al., 2013*; *Scott et al., 2016*). The neddylation of Cul1-Roc1 was also performed as described previously (*Duda et al., 2008*). GST-p130$^{wt}$ (residues 593–790) and GST-p130$^{AA}$ (residues 593–790 R680A L682A) fusions were expressed in *E. coli* BL21-CodonPlus (DE3)-RIL, and then purified by glutathione-affinity and size-exclusion chromatography.

For Cyclin F$^{25-546}$, 1.2 L of Sf9 cells were infected with baculovirus at a density of $2\times10^6$ cells/ml of culture and harvested after 3 days. Pelleted cells were resuspended in lysis buffer (25 mM Tris, 200 mM NaCl, 1 mM DTT, 1 mM PMSF, 1× protease inhibitor cocktail, pH 8.0) and passed through a C3 emulsiflex homogenizer (AVESTIN, Inc). After lysate clarification, the supernatant was loaded onto a GS4B glutathione agarose (Cytvia) column. The column was washed in lysate buffer lacking protease inhibitors and eluted in the same buffer with 10 mM glutathione. Purified TEV protease (1% by mass) was added overnight while protein was dialyzed in buffer containing 25 mM Tris, 200 mM NaCl, and 1 mM DTT (pH 8.0). Protein was then passed back through a GS4B column to remove free GST, concentrated, and loaded in 1 mL onto a Superdex 200 column (Cytvia) equilibrated in similar buffer. Peak fractions from the column elution were pooled and protein concentrated to 8 µM.

Full-length Skp1 was expressed in *E. coli* as a GST fusion protein. 6 L of BL21(DE3) cells were grown to OD of 0.6 then induced with 1 mM IPTG (Isopropyl β-D-1-thiogalactopyranoside) for 16 hr at room temperature. Protein was purified similar to as described above for cyclin F$^{25–546}$, except that following the initial GS4B elution, protein was further purified by Source Q Sepharose ion-exchange chromatography prior to TEV cleavage. For the fluorescence polarization anisotropy assays, the GST tag was left on the cyclin F for stability, and the fusion protein was mixed with Skp1 prior to the experiment.

## P130-cyclin F in vitro binding assays

GST pulldowns were performed by loading 1 µg of purified GST or GST-p130 onto 15 µl of glutathione beads (GoldBio) in NETN (20 mM Tris pH8.0, 100 mM NaCl, 0.5 mM EDTA, 0.5% NP40) for 1 hr, rotating at 4°C. Loaded beads were washed 3× for 5 min in NETN, then incubated with 0.5 mg of whole-cell extract of HEK293T cells transiently transfected with a FLAG-Cyclin F plasmid for 2 hr, rotating, at 4°C. Beads were again washed 3× for 5 min in NETN and p130-cyclin F interaction was assessed by immunoblot.

FLAG pulldowns were performed by immunoprecipitating FLAG-Cyclin F from whole-cell extracts of HEK293T cells transiently transfected with FLAG-Cyclin F. IP was performed by incubating 1 mg whole-cell extract with 50 µl EZView Red anti-FLAG M2 affinity gel (MilliporeSigma) for 2 hr, rotating, at 4°C. Loaded affinity gel was washed 3× for 5 min with NETN, then incubated with 1 µg purified GST or GST-p130 for 2 hr, rotating, at 4°C. Loaded affinity gel was again washed 3× for 5 min in NETN and p130-cyclin F interaction was assessed by immunoblot.

## In vitro ubiquitination assay

Multiple turnover reactions were set up by combining 0.1 µM UBA1, 5 mM MgATP, 0.8 µM CDC34B, 0.3 µM ARIH1, 0.8 µM UBCH7, 0.4 µM NEDD8 to Cul1-Roc1,0.4 µM SKP1, 0.4 µM Cyclin F$^{25–546}$, and 0.6 µM of either GST-p130 or GST-p130$^{AA}$ in the assay buffer (20 mM HEPES pH = 8, 200 mM NaCl) while kept on ice. The mixtures were then equilibrated to room temperature and the reaction was started by the addition of 100 µM of ubiquitin. Reactions were quenched by adding SDS loading buffer at specified time points. After SDS-PAGE, the ubiquitination of GST-p130 was monitored by western blot analysis using α-GST antibody (SC-138, mouse) and a fluorescent secondary antibody (goat anti-mouse IgG (H + L), Alexa Fluor 633, A-21052). Fluorescence was observed using the Amersham Typhoon 5.

## De-phosphorylation assay with calf intestinal phosphatase

Cells were lysed as described above in NETN lacking EDTA. 20 µg total protein lysates were incubated at 37°C for 1 hr with 1× NEB 2.1 buffer (NEB) with 1 unit of calf intestinal phosphatase (NEB) or buffer as a negative control. As a positive control, 1 mM sodium orthovanadate was added to a reaction containing calf intestinal phosphatase for the duration of the 1-hr incubation. Phosphorylation was assessed by immunoblotting.

## Proliferation assays

### Cell counting:

NHF-1 inducible p130 cells were seeded in 10 cm plates, and p130 WT or AA was induced by the addition of 100 ng/ml doxycycline (or DMSO as a control) to media. Cells were trypsinized (Gibco), counted with an automatic cell counter (Bio-Rad), and replated in fresh doxycycline-containing media every 48 hr for 2 weeks. Cell count was plotted versus time, differences were assessed using a Student's t-test, and cell count was used to calculate doubling times. Experiments were carried out in triplicate and repeated three times.

### PrestoBlue:

NHF-1 inducible p130 cells were seeded in 96-well plates, and p130 WT or AA was induced by the addition of 100 ng/ml doxycycline to media. Media were replaced with fresh doxycycline-containing media every 48 hr for 2 weeks, and growth was assessed with the PrestoBlue cell viability reagent (Thermo Fisher Scientific) on days 1, 7, and 14. Experiments were carried out with six technical replicates, and repeated three times.

### Crystal violet assay:

NHF-1 inducible p130 cells were seeded in 60 mm plates, and p130 WT or AA was induced by the addition of 100 ng/ml doxycycline (or DMSO as a control). Media were replaced with fresh doxycycline-containing media (or DMSO-containing media as a control) every 48 hr for 10 days. On day 10, confluence was visualized by staining with Crystal Violet Staining Buffer (0.5% crystal violet, 20% methanol) for 15 min at room temperature, and de-staining with water. Experiments were repeated three times.

## RT-qPCR

In NHF-1 inducible p130 cells, p130 WT or AA was induced by the addition of 100 ng/ml doxycycline (or DMSO as a control) for 14 days. Cells were harvested and RNA was extracted using the RNeasy Plus Mini Kit (QIAGEN). 1 µg of extracted RNA was used to generate cDNA libraries using the SuperScript III First-Strand synthesis system (Thermo Fisher Scientific) following the manufacturer's instructions. Samples were diluted 1:10, except for GAPDH analysis that were diluted 1:1000. Transcript abundance was quantified using the SSO Advanced Universal SYBR Green Supermix (Bio-Rad) and measured with a QuantStudio 6 Flex Real-Time PCR System (Thermo Fisher Scientific), and transcript levels were normalized to GAPDH. Relative quantity of transcripts was quantified using the 2-ΔΔCT method. Each sample was run in triplicate. Primers are listed in tables provided in Appendix 1.

## Flow cytometry

In NHF-1 inducible p130 cells, p130 WT or AA was induced by the addition of 100 ng/ml doxycycline (or DMSO as a control) for the indicated number of days. For cell cycle analysis: 30 min prior to fixation, cells were pulsed with 10 µM EdU and were fixed in 4% formaldehyde/PBS for 15 min at room temperature. Cells were pelleted and re-suspended in 1% BSA/PBS. EdU was labeled with Alexa Fluor 488 using click chemistry as previously described (*Franks et al., 2020*). For Annexin V apoptosis/necrosis: the dead cell apoptosis kit was used following manufacturer instructions (Thermo Fisher Scientific). Flow was carried out on an Attune NxT Flow Cytometer (Thermo Fisher Scientific), and data were analyzed using FlowJo software.

## Florescence polarization anisotropy assay

For the GST-cyclin F$^{25-546}$-Skp1 titration, TAMRA-labeled synthetic peptide (E2F1$^{84-99}$ or p130$^{674-692}$) at 10 nM was mixed with varying concentrations of protein complex in a buffer containing 25 mM Tris,

150 mM NaCl, 1 mM DTT, 0.1% (v/v) Tween-20, pH 8.0. For the $K_i$ measurements, varying concentrations of GST-p130$^{593-790}$ WT, GST-p130$^{593-790}$ R680A/L682A mutant, or synthetic unlabeled E2F1$^{84-99}$ or p130$^{674-692}$ peptide were mixed with 10 nM TAMRA-labeled E2F1$^{84-99}$ and 0.5 µM GST-cyclin F-Skp1 in a buffer containing 25 mM Tris, 150 mM NaCl, 1 mM DTT, 0.1% (v/v) Tween-20, pH 8.0. 40 µl of the reaction were used for the measurement in a 384-well plate. Fluorescence anisotropy (FA) measurements were made in triplicate, using a PerkinElmer EnVision plate reader. The $K_D$ and $K_i$ values were calculated using Prism 8 (Version 8.4.3).

## Iterative indirect immunofluorescence imaging (4i)

Cells were plated in glass-bottom plates (Cellvis) treated as required and prepared as follows. In between each step, samples were rinsed 3× with phosphate-buffered saline (PBS) and incubations were at room temperature, unless otherwise stated. Cells were fixed with 4% paraformaldehyde (Thermo Fisher Scientific, 28908) for 30 min, permeabilized with 0.1% Triton X-100 in PBS for 15 min and inspected for sample quality control following Hoechst staining (1:1000, MilliporeSigma, 99403) in imaging buffer (IB: 700 mM N-acetyl-cysteine (Sigma-Aldrich, A7250) in ddH$_2$O. Adjust to pH 7.4). Cells were rinsed 3× with ddH$_2$O and incubated with elution buffer (EB: 0.5 M L-Glycine (Sigma-Aldrich, 50046)), 3 M Urea (Sigma-Aldrich, U4883), 3 M Guanidine chloride (Thermo Fisher Scientific, 15502-016), and 70 mM TCEP-HCl (Sigma-Aldrich, 646547) in ddH$_2$O. Adjusted to (pH 2.5) 3× for 10 min on shaker to remove Hoechst stain. Sample was incubated with 4i blocking solution (sBS: 100 mM maleimide [Sigma-Aldrich, 129585], 100 mM NH$_4$Cl [Sigma-Aldrich, A9434], and 1% bovine serum albumin in PBS) for 1 hr and incubated with primary antibodies diluted as required (anti-phospho-RB [S807/S811] [1:1000, Cell Signaling Technology, 8516]; anti-RB [1:500, Cell Signaling Technology, 9309]; anti-p21 [1:200, R&D Systems, AF1047]; anti-p130 [1:100, Cell Signaling Technology, 13610]; anti-phospho-H2A.X [Ser139] [1:200, Cell Signaling Technology, 80312], anti-phospho-CHK1 [S317] [1:800, Cell Signaling Technology, 12302], anti-53BP1 [1:250, Abcam, ab36823], CDT1 [1:200, Cell Signaling Technology, 8064], CDC6 [1:100, Santa Cruz, sc-9964], HA [1:200, BioLegend, 901501]) in conventional blocking solution (cBS: 1% bovine serum albumin in PBS) overnight at 4°C. Samples were rinsed 3× with PBS and then incubated in secondary antibodies (1:500, donkey anti-rabbit AlexaFluor Plus 488 [Thermo Fisher Scientific, A32790], donkey anti-mouse AlexaFluor Plus 555 [Thermo Fisher Scientific, A32773], and donkey anti-goat AlexaFluor Plus 647 [Thermo Fisher Scientific, A32758]) and Hoechst for 1 hr on shaker, then rinsed 5× with PBS and imaged in IB. Samples were imaged using the Nikon Ti Eclipse inverted microscope with a Nikon Plan Apochromat Lambda 40× objective with a numerical aperture of 0.95 and an Andor Zyla 4.2 sCMOS detector. Stitched 8×8 images were acquired for each condition using the following filter cubes (Chroma): DAPI(383-408/425/435-485nm), GFP(450-490/495/500-550nm), Cy3(530-560/570/573-648nm), and Cy5(590-650/660/663-738nm). After imaging, samples were rinsed 3× with ddH2O, antibodies were eluted, and re-stained iteratively as described above.

## 4i image analysis

Nuclear segmentation and quantification were performed using standard modules in CellProfiler (v3.1.8) as described below. For each round of immunofluorescence images obtained by 4i, individual nuclei were automatically detected and segmented using the IdentifyPrimaryObjects module. Cytoplasmic segmentation was performed by making a 5-pixel ring outside the nucleus using the ExpandOrShrinkObjects and IdentifyTertiaryObjects modules. Nuclear and cytoplasmic intensities were quantified using the MeasureObjectIntensity module. Distributions of single-cell intensities were visualized using the seaborn library (v0.11.0) in Python (v3.7.1).

## Apoptosis assays

Cells were grown for 0–14 days in 100 ng/ml doxycycline or water as a vehicle control. As a positive control for apoptosis. As a positive control, cells were treated with 100 nM staurosporine for 6 hr to induce apoptosis. Presence of cleaved PARP and cleaved Caspase three was assessed by immunoblot. And, apoptosis/necrosis were assessed using an Annexin V/propidium iodide staining kit (Thermo Fisher Scientific) coupled with flow cytometry.

## Quantification and statistical analysis

Western blot quantification was carried out using the Fiji software. All statistical analyses were carried out using GraphPad Prism v9.

## Acknowledgements

The authors thank lab members for helpful discussions throughout this project. The authors thank Brenda Schulman (Max Planck Institute of Biochemistry) for generously providing SCF complex reagents used for in vitro ubiquitination assays. The authors thank Larisa Litovchick (Virginia Commonwealth University) for providing the pBABE-p130 vector. The authors thank Dennis Goldfarb (Washington University) for help with DepMap data downloads and analysis. The Emanuele lab (TPE and MJE) is supported by the UNC University Cancer Research Fund (UCRF), the National Institutes of Health (R01GM120309, R01GM134231), and the America Cancer Society (Research Scholar Grant; RSG-18-220-01-TBG). The Brown lab (ETW and NGB) is supported by UCRF and National Institutes of Health (R35GM128855) and ETW is partially supported by R01CA163834. The Rubin lab (PN and SMR) is supported by the National Institutes of Health (R01GM127707). The Purvis lab (WMS and JEP) is supported by by NIH grants R01-GM138834 (JEP), DP2-HD091800 (JEP), and NSF CAREER Award 1845796 (JEP).

## Additional information

### Funding

| Funder | Grant reference number | Author |
| --- | --- | --- |
| National Institute of General Medical Sciences | R01GM120309 | Taylor P Enrico Xianxi Wang Michael J Emanuele |
| National Institute of General Medical Sciences | R01GM134231 | Taylor P Enrico Xianxi Wang Michael J Emanuele |
| National Institute of General Medical Sciences | R35GM128855 | Elizaveta T Wick Nicholas G Brown |
| American Cancer Society | RSG-18-220-01-TBG | Taylor P Enrico Xianxi Wang Michael J Emanuele |
| National Cancer Institute | R01CA163834 | Elizaveta T Wick |
| National Institute of General Medical Sciences | R01GM127707 | Peter Ngoi Seth M Rubin |
| National Institute of General Medical Sciences | GM138834 | Wayne Stallaert Jeremy E Purvis |
| National Institute of General Medical Sciences | DP2-HD091800 | Wayne Stallaert Jeremy E Purvis |
| National Science Foundation | 1845796 | Jeremy E Purvis |
| National Institute of General Medical Sciences | T32 GM007040 | Taylor P Enrico |
| National Institute of General Medical Sciences | T32 GM007040 | Taylor P Enrico |

The funders had no role in study design, data collection and interpretation, or the decision to submit the work for publication.

### Author contributions

Taylor P Enrico, Conceptualization, Data curation, Formal analysis, Investigation, Methodology, Validation, Visualization, Writing - original draft, Writing – review and editing; Wayne Stallaert, Peter Ngoi, Formal analysis, Investigation, Methodology, Visualization, Writing – review and editing; Elizaveta T Wick, Investigation, Methodology, Writing - original draft, Writing – review and editing; Xianxi Wang, Investigation, Methodology; Seth M Rubin, Nicholas G Brown, Jeremy E Purvis, Conceptualization, Formal analysis, Funding acquisition, Methodology, Project administration, Supervision, Visualization,

Writing – review and editing; Michael J Emanuele, Conceptualization, Funding acquisition, Project administration, Supervision, Visualization, Writing - original draft, Writing – review and editing

### Author ORCIDs
Taylor P Enrico 
Seth M Rubin 
Jeremy E Purvis 
Michael J Emanuele 

### Decision letter and Author response
Decision letter https://doi.org/10.7554/eLife.70691.sa1
Author response https://doi.org/10.7554/eLife.70691.sa2

---

## Additional files

### Supplementary files
• Transparent reporting form

• Source code 1. Violin plotting.

• Source data 1. Immunoblots. This source data file includes all uncropped blots used to generate data for the main figures and figure supplements. Additionally, copies of the uncropped images are shown a second time where blot strips shown in figures are highlighted with a red square and the protein that was blotted for is noted.

• Source data 2. Cycloheximide Chase. This source data file contains quantified band densities and background used to generate the cycloheximide protein degradation plot and for the protein half-life calculations.

• Source data 3. Flow cytometry. This source data file contains percentages of cells in various populations for the flow cytometry experiments.

• Source data 4. Cell counting and PrestoBlue. This source data file contains the raw cell counts for the courting experiment and the raw fluorescence measurements for the PrestoBlue experiment.

• Source data 5. Rt-qPCR. This source data file contains the output from the rt-qPCR machine that was used to determine gene expression for the rt-qPCR blot.

### Data availability
Unprocessed, uncropped, immunoblots are made available in the supplemental source data. All raw, unprocessed imaging data is available at Dryad. Raw data related to cell proliferation assays (cell counting and Presto-blue analysis), RT-qPCR, immunoblot quantification for cycloheximide chase experiments and flow cytometry is available in the supplemental source data. All reagents related to this work will be made fully available upon request.

The following dataset was generated:

| Author(s) | Year | Dataset title | Dataset URL | Database and Identifier |
|---|---|---|---|---|
| Enrico T, Stallaert W, Wick E, Ngoi P, Emanuele M, Rubin S, Brown N, Purvis J | 2021 | Data from: Cyclin F drives proliferation through SCF-dependent degradation of the retinoblastoma-like tumor suppressor p130/RBL2 | https://doi.org/10.5061/dryad.69p8cz93d | Dryad Digital Repository, 10.5061/dryad.69p8cz93d |

The following previously published datasets were used:

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

## Appendix 1

## Key reagents table
Lists key reagents used in this study.

**Appendix 1—key resources table**

| Reagent type (species) or resource | Designation | Source or reference | Identifiers | Additional information |
|---|---|---|---|---|
| Gene (*Homo sapiens*) | CCNF | GenBank | Gene ID: 899 | |
| Gene (*H. sapiens*) | RBL2 | GenBank | Gene ID: 5934 | |
| Antibody | CDK4 (rabbit monoclonal) antibody | CST | Cat. #: D9GE3; RRID:AB_2799229 | IB (1:1000) |
| Antibody | CDK6 (rabbit monoclonal) antibody | CST | Cat. #:D4S8S; RRID:AB_2721897 | IB (1:1000) |
| Antibody | Cyclin A1 (rabbit monoclonal) antibody | Abcam | Cat. #: ab53699; RRID:AB_879763 | IB (1:1000) |
| Antibody | Cyclin A1 (mouse monoclonal) antibody | Santa Cruz | Cat. #: SC-751; RRID:AB_631329 | IB (1:5000); IP (1 µg) |
| Antibody | Cyclin D1 (rabbit monoclonal) antibody | CST | Cat. #: 2978; RRID:AB_2259616 | IB (1:1000) |
| Antibody | Cyclin E1 (mouse monoclonal) antibody | CST | Cat. #: 4129; RRID:AB_2071200 | IB (1:5000); IP (1 µg) |
| Antibody | Cyclin E1 (rabbit monoclonal) antibody | CST | Cat. #: 20808; RRID:AB_2783554 | IB (1:1000) |
| Antibody | Cyclin F (rabbit polyclonal) antibody | Santa Cruz | Cat. #: SC-952; RRID:AB_2071212 | IB (1:5000) |
| Antibody | FLAG (HRP-conjugated mouse monoclonal) antibody | Sigma-Aldrich | Cat. #: A8592; RRID:AB_439702 | IB (1:10,000) |
| Antibody | GAPDH (mouse monoclonal) antibody | Santa Cruz | Cat. #: sc-47724; RRID:AB_627678 | IB (1:5000) |
| Antibody | GST (HRP-conjugated mouse monoclonal) antibody | GeneTex | Cat. #: GTX114099; RRID:AB_1949436 | IB (1:5000) |
| Antibody | HA (mouse monoclonal) antibody | Covance/ BioLegend | Covance Cat. #: MMS-101P; RRID:AB_2314672 | IB (1:2000) |
| Antibody | Lin54 (rabbit polyclonal) antibody | Bethyl | Cat. #: A303-799A; RRID:AB_11218173 | IB (1:1000) |
| Antibody | Normal rabbit IgG (polyclonal) antibody | ProteinTech | Cat. #: 30000-0-AP; RRID:AB_2819035 | IP (1 µg) |
| Antibody | p107 (rabbit monoclonal) antibody | CST | Cat. #: D3P3C; RRID:AB_2800144 | IB (1:1000) |
| Antibody | p130 (rabbit monoclonal) antibody | CST | Cat. #: D9T7M; RRID:AB_2798274 | IB (1:1000) |
| Antibody | p130 pS672 (rabbit monoclonal) antibody | Abcam | Cat. #: ab76255; RRID:AB_2284799 | IB (1:5000) |
| Antibody | p27 (rabbit monoclonal) antibody | CST | Cat. #: 2552; RRID:AB_10693314 | IB (1:1000) |
| Antibody | Skp2 (rabbit monoclonal) antibody | CST | Cat. #: 2652; RRID:AB_11178941 | IB (1:5000) |
| Antibody | Tubulin (mouse monoclonal) antibody | Santa Cruz | Cat. #: 32293; RRID:AB_628412 | IB (1:5000) |

*Appendix 1 Continued on next page*

*Appendix 1 Continued*

| Reagent type (species) or resource | Designation | Source or reference | Identifiers | Additional information |
|---|---|---|---|---|
| Antibody | Goat anti-mouse IgG HRP-conjugated (goat polyclonal) antibody | Jackson ImmunoResearch | Cat. #: 115-035-003; RRID:AB_10015289 | IB (1:5000) |
| Antibody | Goat anti-rabbit IgG HRP-conjugated (goat polyclonal) antibody | Jackson ImmunoResearch | Cat. #: 111-035-003; RRID:AB_2313567 | IB (1:5000) |
| Antibody | Phospho-RB(S807/S811) (rabbit monoclonal) antibody | CST | Cat. #: 8516; RRID:AB_11178658 | 4i (1:1000) |
| Antibody | RB (mouse monoclonal) antibody | CST | Cat. #: 9309; RRID:AB_823629 | 4i (1:500) |
| Antibody | p21 (goat polyclonal) antibody | R&D Systems | Cat. #: AF1047; RRID:AB_2244704 | 4i (1:200) |
| Antibody | p130 (rabbit monoclonal) antibody | CST | Cat. #: 13610; RRID:AB_2798274 | 4i (1:100) |
| Antibody | Phospho-H2A.X(ser139) (mouse monoclonal) antibody | CST | Cat. #: 80312; RRID:AB_2799949 | 4i (1:200) |
| Antibody | Anti-phospho-Chk1 (rabbit monoclonal) antibody | CST | Cat. #: 12302; RRID:AB_2783865 | 4i (1:800) |
| Antibody | 53 BP1 (rabbit polyclonal) antibody | Abcam | Cat. #: ab36823; RRID:AB_722497 | 4i (1:250) |
| Antibody | CDT1 (rabbit monoclonal) antibody | CST | Cat. #: 8064; RRID:AB_10896851 | 4i (1:200) |
| Antibody | CDC6 (mouse monoclonal) antibody | Santa Cruz | Cat. #: sc-9964; RRID:AB_627236 | 4i (1:100) |
| Antibody | Donkey anti-rabbit AlexaFluor Plus 488 (Donkey polyclonal) antibody | Thermo Fisher Scientific | Cat. #: A32790; RRID:AB_2762833 | 4i (1:500) |
| Antibody | Donkey anti-goat AlexaFluor plus 647 (Donkey polyclonal) antibody | Thermo Fisher Scientific | Cat. #: A32758; RRID:AB_2762828 | 4i (1:500) |
| Antibody | Donkey anti-mouse AlexaFluor Plus 555 (Donkey polyclonal) antibody | Thermo Fisher Scientific | Cat. #: A32773; RRID:AB_2762848 | 4i (1:500) |
| Strain, strain background (*Escherichia coli*) | BL21 Competent *Escherishia coli* | NEB | Cat. #: C2530H | |
| Strain, strain background (*E. coli*) | DH5-α Competent *Escherishia coli* | NEB | Cat. #: C2988J | |
| Commercial Assay or Kit | GeneJET Plasmid Miniprep Kit | Thermo Fisher Scientific | Cat. #: K0503 | |
| Commercial Assay or Kit | Q5 Site-Directed Mutagenesis Kit (without competent cells) | NEB | Cat. #: E0552S | |
| Commercial Assay or Kit | Rneasy Plus Mini Kit | QIAGEN | Cat. #: 74134 | |
| Commercial Assay or Kit | SuperScript III First-Strand Synthesis System | Thermo Fisher Scientific | Cat. #: 18080051 | |

*Appendix 1 Continued on next page*

*Appendix 1 Continued*

| Reagent type (species) or resource | Designation | Source or reference | Identifiers | Additional information |
|---|---|---|---|---|
| Commercial Assay or Kit | Dead Cell Apoptosis Kit with Annexin V FITC and PI for flow cytometry | Thermo Fisher Scientific | Cat. #: V13242 | |
| Chemical compound, drug | SSO Advanced Universal SYBR Green Supermix | Bio-Rad | Cat. #: 1725271 | |
| Chemical compound, drug | Bio-Rad Protein Assay Dye Reagent Concentrate | Bio-Rad | Cat. #: 5000006 | |
| Chemical compound, drug | PrestoBlue Cell Viability Reagent | Thermo Fisher Scientific | Cat. #: A13261 | |
| Chemical compound, drug | Calf Intestinal Phosphatase (CIP) | NEB | Cat. #: M0290 | |
| Chemical compound, drug | Clarity ECL Western Blot Substrate | Bio-Rad | Cat. #: 1705060 | |
| Chemical compound, drug | Cyclohexamide | MilliporeSigma | Cat. #: 1810 | |
| Chemical compound, drug | MG132 | Selleck Chemicals | Cat. #: S2619 | |
| Chemical compound, drug | MLN4924 | Active Biochem | Cat. #: A-1139 | |
| Chemical compound, drug | Glutathione Agarose Resin | GoldBio | Cat. #: G-250-5 | |
| Chemical compound, drug | EZView Red Anti-FLAG M2 Affinity Gel | MilliporeSigma | Cat. #: F2426 | |
| Chemical compound, drug | EZView Red Anti-HA Affinity Gel | MilliporeSigma | Cat. #: E6779 | |
| Chemical compound, drug | PierceTM Protein A/G agarose Beads | Thermo Fisher Scientific | Cat. #: 20421 | |
| Chemical compound, drug | Rnase A | MilliporeSigma | Cat. #: R6513 | |
| Chemical compound, drug | Lipofectamine RNAiMAX | Thermo Fisher Scientific | Cat. #: 13778150 | |
| Chemical compound, drug | PolyJet Transfection Reagent | SignaGen | Cat. #: SL100688 | |
| Chemical compound, drug | Lipofectamine 2000 | Thermo Fisher Scientific | Cat. #: 11668019 | |
| Cell line (*H. sapiens*) | 293T | ATCC | Cat. #: CRL-3216, RRID:CVCL_0063 | |
| Cell line (*H. sapiens*) | U2OS | ATCC | Cat. #: HTB-96; RRID:CVCL_0042 | |
| Cell line (*H. sapiens*) | HeLa | ATCC | Cat. #: CCL-2; RRID:CVCL_0030 | |
| Cell line (*H. sapiens*) | HeLa sgCTRL | PMID: 27653696 | | |
| Cell line (*H. sapiens*) | HeLa sgCCNF | PMID: 27653696 | | |
| Cell line (*H. sapiens*) | MCF7 pIND CCNF | This paper | | See *Figure 1*, *Figure 1—figure supplement 3* |
| Cell line (*H. sapiens*) | T47D pIND CCNF | This paper | | See *Figure 1*, *Figure 1—figure supplement 3* |
| Cell line (*H. sapiens*) | NHF-1 | William Kaufman Lab (UNC; retired) | | |

*Appendix 1 Continued on next page*

*Appendix 1 Continued*

| Reagent type (species) or resource | Designation | Source or reference | Identifiers | Additional information |
|---|---|---|---|---|
| Cell line (*H. sapiens*) | IMR-90 | Yue Xiong Lab (UNC; retired) | | |
| Cell line (*H. sapiens*) | T98G | Tissue Culture Facility, UNC | | |
| Cell line (*H. sapiens*) | NHF-1 doxy-inducible p130 WT | This paper | | See *Figure 6*, *Figure 7*, *Figure 5—figure supplements 1 and 2* |
| Cell line (*H. sapiens*) | NHF-1 doxy-inducible p130 AA | This paper | | See *Figure 6*, *Figure 5—figure supplements 1 and 2* |
| Transfected Construct (*H. sapiens*) | pBABE-p130 | Gift from Larisa Litovchick lab (VCU) | | For lentiviral transfection |
| Transfected Construct (*H. sapiens*) | pDEST-HA3-p130 | This paper | | Transfected construct (human); See *Figures 3–5*, *Figure 4—figure supplement 1* |
| Transfected Construct (*H. sapiens*) | pDEST-HA3-p130 1–417 | This paper | | Transfected construct (human); See *Figure 4—figure supplement 1* |
| Transfected Construct (*H. sapiens*) | pDEST-HA3-p130 418–1139 | This paper | | Transfected construct (human); See *Figure 4—figure supplement 1* |
| Transfected Construct (*H. sapiens*) | pDEST-HA3-p130 418-616 | This paper | | Transfected construct (human); See *Figure 4—figure supplement 1* |
| Transfected Construct (*H. sapiens*) | pDEST-HA3-p130 418-827 | This paper | | Transfected construct (human); See *Figure 4—figure supplement 1* |
| Transfected Construct (*H. sapiens*) | pDEST-HA3-p130 418-1024 | This paper | | Transfected construct (human); See *Figure 4—figure supplement 1* |
| Transfected Construct (*H. sapiens*) | pDEST-HA3-p130 828-1139 | This paper | | Transfected construct (human); See *Figure 4—figure supplement 1* |
| Transfected Construct (*H. sapiens*) | pDEST-HA3-p130 R658A I660A | This paper | | Transfected construct (human); See *Figure 4—figure supplement 1* |
| Transfected Construct (*H. sapiens*) | pDEST-HA3-p130 R680A L682A | This paper | | Transfected construct (human); See *Figure 4—figure supplement 1* |
| Transfected Construct (*H. sapiens*) | pDEST-FLAG-Cyclin F | PMID: 27653696 | | Transfected construct (human) |
| Transfected Construct (*H. sapiens*) | Cyclin F M309A L131A | PMID: 20596027 | | Transfected construct (human) |
| Transfected Construct (*H. sapiens*) | pDEST-FLAG-Cyclin F M309A L313A | This paper | | Transfected construct (human) |
| Transfected Construct (*H. sapiens*) | pGEX-GST-p130 593-790 | PMID: 9188854 | | For protein expression in *Escherichia coli* |
| Transfected Construct (*H. sapiens*) | pGEX-GST-p130 593-790 R680A L682A | This paper | | For protein expression in *Escherichia coli* |

*Appendix 1 Continued on next page*

*Appendix 1 Continued*

| Reagent type (species) or resource | Designation | Source or reference | Identifiers | Additional information |
|---|---|---|---|---|
| Transfected Construct (*H. sapiens*) | pINDUCER20 | PMID: 21307310 | Addgene #44012; RRID:Addgene_44012 | For lentiviral transfection |
| Transfected Construct (*H. sapiens*) | pINDUCER20 CCNF | This paper | | For lentiviral transfection |
| Transfected Construct (*H. sapiens*) | pLV[Exp]-CMV> Tet3G/ Hygro | VectorBuilder | ID:VB180123-1018bxq | For lentiviral transfection |
| Transfected Construct (*H. sapiens*) | pLV[TetOn]-Neo-TRE3G > HA/{p130 AA} | this paper, VectorBuilder | ID:VB200319-6469hpy | For lentiviral transfection |
| Transfected Construct (*H. sapiens*) | pLV[TetOn]-Neo-TRE3G > HA/{p130 WT} | this paper, VectorBuilder | ID:VB200319-6451nqd | For lentiviral transfection |
| Sequence-based reagent | p130 from AA1 w/attb site Forward | PCR primers | | 5'-GGGGACAAGTT TGTACAAAAAAG CAGGCTTAAT GCCGTCGGGAGGTGACCAG |
| Sequence-based reagent | p130 from AA417 w/attb site Reverse | PCR primers | | 5'-GGGGACCACTTTGTACAA GAAAGCTGGGTA CTACACACAAG GGCTATTCTCCTT |
| Sequence-based reagent | p130 from AA418 w/attb site Forward | PCR primers | | 5'-GGGGACAAGTTTGTA CAAAAAAGCAGGCTT AACTCCAGTTTCTACAGCTACG |
| Sequence-based reagent | p130 from AA 1139 w/attb site Reverse | PCR primers | | 5'-GGGGACCACTTTGTACAA GAAAGCTGGGTA TCAGTGG GAACCACGGTCATT |
| Sequence-based reagent | p130 from AA 616 w/attb site Reverse | PCR primers | | 5'-GGGGACCACTTTGTACAAG AAAGCTGGG TACTAAACTCTGTTT TCATTGTCTCT |
| Sequence-based reagent | p130 from AA 827 w/attb site Reverse | PCR primers | | 5'-GGGGACCACTTTGTA CAAGAAAGCTGGG TACTAACTACTGCT GGTTACAGACTG |
| Sequence-based reagent | p130 from AA 1024 w/attb site Reverse | PCR primers | | 5'-GGGGACCACTTTGTACAA GAAAGCTGGG TACTAGTACTTCA TGGCAAATGTCTT |
| Sequence-based reagent | p130 from AA 828 w/attb site Forward | PCR primers | | 5'-GGGGACAAGTTTGTAC AAAAAAGCAGGCTT AAATAGACCCAGGAAGACCAGC |
| Sequence-based reagent | p130 R658A I660A Forward | PCR primers | | 5'-CGCCACATCTCCAA CCACATTATAC |
| Sequence-based reagent | p130 R658A I660A Reverse | PCR primers | | 5'-CTGGCTCCAAGT CCTCCAGTATC |
| Sequence-based reagent | p130 R680A L682A Forward | PCR primers | | 5'-GGCCTTTGTTGAGAA TGATAGCCCCTC |
| Sequence-based reagent | p130 R680A L682A Reverse | PCR primers | | 5'-CGGGCTCTGGTAG TGCTGGCTGG |
| Sequence-based reagent | Cyclin F from AA 1 w/attb site Forward | PCR primers | | 5'-GGGGACAAGTTTGTACA AAAAGCAGGC TTAATGGGGAGCGGCGG CGTGGTCC |

*Appendix 1 Continued on next page*

| Reagent type (species) or resource | Designation | Source or reference | Identifiers | Additional information |
|---|---|---|---|---|
| Sequence-based reagent | Cyclin F from end w/attb site Reverse | PCR primers | | 5'-GGGGACCACTTT GTACAAGAAAGCTGG GTATTACAGCCTCA CAAGGCCCAGG |
| Sequence-based reagent | CCNE1 RT-qPCR Forward | PCR primers | | 5'-AGACATACTTAA GGGATCAGC |
| Sequence-based reagent | CCNE1 RT-qPCR Reverse | PCR primers | | 5'-CACACCTCCATTA ACCAATC |
| Sequence-based reagent | CDC6 RT-qPCR Forward | PCR primers | | 5'-ATGTAAATCACC TTCTGAGC |
| Sequence-based reagent | CDC6 RT-qPCR Reverse | PCR primers | | 5'-GTCATCCTGTT ACCATCAAC |
| Sequence-based reagent | DHFR RT-qPCR Forward | PCR primers | | 5'-TTCCAGAAGTCT AGATGATGC |
| Sequence-based reagent | DHFR RT-qPCR Reverse | PCR primers | | 5'-CTTCCTTATAAACAGAA CTGCC |
| Sequence-based reagent | E2F1 RT-qPCR Forward | PCR primers | | 5'-CTGATGAATATCTGTA CTACGC |
| Sequence-based reagent | E2F1 RT-qPCR Reverse | PCR primers | | 5'-CTTTGATCACCATAAC CATCTG |
| Sequence-based reagent | GAPDH RT-qPCR Forward | PCR primers | | 5'-GGCCTCCAAGGA GTAAGACC |
| Sequence-based reagent | GAPDH RT-qPCR Reverse | PCR primers | | 5'-AGGGGTCTACA TGGCAACTG |
| Sequence-based reagent | Cyclin F RT-qPCR Forward | PCR primers | | 5'-AGGACAAGC GCTATGGAGAA |
| Sequence-based reagent | Cyclin F RT-qPCR Reverse | PCR primers | | 5'-TCTGTCTTCC TGGAGGCTGT |
| Sequence-based reagent | CDT1 RT-qPCR Forward | PCR primers | | 5'-CCTGGGGAAATGGAGAAG |
| Sequence-based reagent | CDT1 RT-qPCR Reverse | PCR primers | | 5'-TTGTCCAGCTTGACGTAG |
| Sequence-based reagent | Cyclin F subcloning into pfastbac Forward | PCR primers | | 5'-GCTAGGGTCGGATCCAG GAGGCCCCGAAACCTGACC |
| Sequence-based reagent | Cyclin F subcloning into pfastbac Reverse | PCR primers | | 3'-GCTAGGCATAGCGGCC GCACCTTAGCTGT CTTGTGTCACT CCTAATGCAGC |
| Sequence-based reagent | Skp1 subcloning into PGEX-4T-1 Forward | PCR primers | | 5'-GCTAGGGTCGGA TCCATGCCTT CAATTAAGTTGCAGAG TTCTGATGG |
| Sequence-based reagent | Skp1 subcloning into PGEX-4T-1 Reverse | PCR primers | | 3'-GCTAGGCATAGCCTC GAGTTACTT CTCTTCAC ACCACTGGTTCTC |
| Sequence-based reagent | CCNF #1 | siRNA | | 5'-UAGCCUACCU CUACAAUGAUU |
| Sequence-based reagent | CCNF #2 | siRNA | | 5'-GCACCCGGU UUAUCAGUAAUU |
| Sequence-based reagent | siFF | siRNA | | 5'-CGUACGCGGA AUACUUCGAUU |
| Other | EdU | Sigma-Aldrich | Cat. #: T511285 | For flow cytometry—10 µM to cell media for 30 min prior to fixation |
| Other | DAPI | Thermo Fisher Scientific | Cat. #: D1306 | For flow cytometry (1 µg/ml) |

| Reagent type (species) or resource | Designation | Source or reference | Identifiers | Additional information |
|---|---|---|---|---|
| Other | Alexa-Fluor 488 Azide | Thermo Fisher Scientific | Cat. #: A10266 | For flow cytometry (0.2 µM) |

