## [Editor Report]

The identification of the tumor suppressor RBL2/p130 as a substrate of cyclin F/SCF adds a new level of understanding about the role of this ubiquitin ligase in cell cycle control and identifies a novel functional interaction that could have implications for cancer. This work will be of interest to researchers in the fields of cell cycle and cancer biology.

---

## [Decision Letter]

**Decision letter after peer review:**

Thank you for submitting your article "Cyclin F drives proliferation through SCF-dependent degradation of the retinoblastoma-like tumor suppressor p130/RBL2" for consideration by *eLife*. Your article has been reviewed by 3 peer reviewers, and the evaluation has been overseen by a Reviewing Editor and Maureen Murphy as the Senior Editor. The following individual involved in review of your submission has agreed to reveal their identity: Michele Pagano (Reviewer #3).

Essential revisions

1) The authors provided evidence that an R680xL682/AxA mutant is unable to interact with cyclin F and that this mutant blocks cell cycle gene expression, cell cycle progression and proliferation. The authors do not acknowledge or account for prior reports that this conserved RxL motif has been implicated in cyclin A and cyclin E binding in both p130 and p107 (Lacy and Whyte, 1997) as well as cyclin D binding in p107 (Leng et al., 2002). Thus, the p130 interaction data and functional data for the AxA mutant are likely to be true and unrelated as the functional data for the AxA likely results from a deficiency in CDK phosphorylation rather than increased stability. The authors need to show that the p130 AxA mutant can still interact with cyclins D, A, and E and be phosphorylated at CDK sites such as S672 for the alternative deficiency in CDK phosphorylation hypothesis to be ruled out for the AxA mutant.

2) If the p130 AxA mutant is competent for CDK phosphorylation, this raises questions regarding the disassembly of the DREAM complex. Previously, CDK4/6 phosphorylation of p130 has implicated in DREAM disassembly but, if the repression of cell cycle gene expression observed in the AxA mutant is due to increased protein stability rather than CDK phosphorylation, this would indicate that p130 degradation is the rate limiting step of DREAM disassembly or that p130 has a transcriptional repression function independent of the DREAM complex. The authors should address p130 degradation within DREAM complex function to resolve this conflict as to reduced p130 degradation leads to increase repression of cell cycle gene expression.

3) Many of the cellular experiments establishing that p130 is cyclin F substrate (Figure 2E, 3, 4B, 5B, S3A) were conducted in 293T cells that express Adenovirus 5 E1A and SV40 Large T antigen which bind and inactivate Rb family members and disrupt the RB-CDK network. Furthermore, SV40 large T promotes dephosphorylation of p130 (Lin and DeCaprio, 2003) explaining why SKP2 overexpression did not alter p130 levels as the SKP2 interacts with phosphorylated p130. SV40 LT also binds directly to FBXW7. Thus, the generalizability of these experiments is limited. Ideally, all the experiments in 293T cells would be repeated in cells where the RB-CDK network was intact but the strong in vitro data for the mapping the cyclin F-p130 interaction on p130 indicates that this may be unnecessary for Figure 2E, 4B, and S3A. At a minimum, the experiments in Figure 3, especially 3B, and 5B should be repeated in cells where the RB-CDK network is intact.

4) The transition from Figure 1 to Figure 2 is a difficult to follow in the text as Figure 2A and 2B are correlative observations. Figure 2A and 2B feel more like a follow up to 1D than addressing the question of whether and SCF cyclin F complex regulates p130 levels. Consequently, please move Figures 2A and 2B to figure 1 and Figure S2 should be brought into the main figure.

5) Figures 6 and 7 are poorly organized and the narrative is a bit difficult to follow and appreciate as the authors bounce between cell cycle, proliferation, and apoptotic readouts. It may strengthen and focus the story to focus on cell cycle gene expression and cell cycle progression in Figure 6 and cellular proliferation and apoptosis in Figure 7. This would also help the reader appreciate the CDK4/6i inhibitor data more as CDK4/6i inhibition is generally considered cytostatic rather than cytotoxic.

6) Since CDT1 protein species are degraded by multiple complexes, please include RT-qPCR data for CDT1 so the reader may appreciate that the decrease in CDT1 levels results from reduced transcription.

7) On lines 293-295, the authors compare p130 levels in S4A and 6A to point out the induced p130 AxA accumulates to higher levels at later time points compared to p130 WT. It is difficult to appreciate this comparison as these are different blots in different figures. Consequently, to make this point, the authors should include an IB as in S4A to show p130 WT and AxA levels over the 14 day time course. Ideally, time points would be collected.

8) The authors should include representative images for key time points and comparison for the multiple figures of quantitative immunofluorescence experiments, particularly for the nuclear to cytoplasmic ratio. Also the authors should more fully explain their analysis workflow as data can be processed through the "standard modules" cell profiler to differing results. If a previously established workflow was used that has been previously described in the literature, please provide a citation for the workflow.

9) Inactivation of p130 by CDK phosphorylation is likely to dominate p130 degradation in functional assay under normal growth conditions. The authors should consider examining the p130 degradation after DNA damage, in presence of reduced serum levels, or during quiescence (by expression a cyclin F KEN box mutant) to discern the functional impact of altered p130 degradation by p130.

10) It would help the reader if the results of the iterative immunofluorescence staining were supplemented by additional data such as representative cell images and/or immunoblots. It is difficult to appreciate the extent and the significance of the observed effects in the format shown in Figures 6F-H, 7B-C and S5C-E.

11) Standard methods such as immunoblotting or protein expression in bacteria, are described in excessive detail, whereas the 4i image analysis that could be helpful to a reader unfamiliar with this technique, is described rather briefly. It took a while and reading some additional papers for this reviewer to understand what in fact was shown in the graphs obtained by this technique.

12) Figure 1. From the description, it appears that results of the DepMap correlation analysis form the premise for this work by showing that depletion of cyclin F has similar impact on the cancer cell fitness as depletion of several components of the Rb-CDK pathway, including p130. The authors then hypothesize that cyclin F may regulate the expression of these factors, which leads to identification of p130 as a cyclin F substrate. Significance of this analysis should be better explained. Indeed, the authors first make the introduction that closely correlated genes in DepMap are often involved in the same protein complex or a functional pathway, and then go on to show that cyclin F and p130 not only have the opposite patterns if expression, but also negatively regulate each other, which would be contradictory to a direct correlation found in DepMap analysis. It would help the reader if the authors explained this apparent contradiction, and ideally validated the conclusions of the DepMap analysis by comparing the effects of cyclin F and p130 knockdown in a panel of select cancer cell lines.

13) Figure 2C: In HeLa cells, all Rb family members are targeted to degradation through the interaction with HPV E7 oncoprotein. Although loss of CCNF appears to increase the levels of p130 in HeLa cells, this result does not strongly support the direct role of CCNF in p130 degradation without ruling out an indirect effect on E7 function. Additionally, loss of CCNF could influence the cell cycle synchronization therefore the cell cycle analysis data should be included with this experiment. In general, since cyclin F is involved in regulation of many cell-cycle related factors, the cell cycle data should be included in the experiments where the levels of CCNF are manipulated such as Figure 2D.

14) Figure 2E: Again, the choice of 293T cell line to demonstrate the binding between the endogenous p130 and cyclin F is surprising. The levels of p130 in these cells are very low because of the presence of SV40 Large T antigen that targets Rb family members for degradation. Furthermore, SV40 LT binds to Rb family members and can indirectly recruit other factors into the complexes with these proteins. Therefore, the endogenous interaction between p130 and cyclin F should be confirmed in another cell line. Figure 3: Same concerns – the presence of SV40 LT could complicate the interpretation of these results. The authors need to confirm these effects in another cell line, in the absence of SV40 LT.

15) Figure 4: Panel B: It would be interesting to know whether R680A/L682A p130 mutant can still bind SV40 Large T or HPV E7 proteins. Panel E: The rationale for using E2F1 displacement for measuring the binding affinity between cyclin F-Skp1 complex and p130 fragments is unclear. Is it possible to show a direct binding between these factors?

16) Figure 5B: It is apparent that p130-AA is still being degraded during the CHX chase, but is unaffected by cyclin F overexpression. Could the proteasome or neddylation inhibitors block this degradation? Is it possible that both Skp2 and cyclin F can degrade p130 through different mechanisms? Panel C: Significance of the differences is not shown.

17) Figure 6: It would be very interesting to see how the p130-AA levels are regulated during the G0/G1 arrest and progression through the cell cycle. This could be a key experiment to demonstrate the role of the 680-RRL site for cell cycle dependent degradation of p130. Panel E: Description of this figure indicates that cells were pulse-labeled with EdU to detect S-phase, but the figure only shows G0/G1 fraction. Changes in S-phase and G2/M fractions could be informative and should be shown. Also, this experiment needs to be better explained. Were the cells allowed to grow to confluency over 14 days? Was the growth media changed or supplemented throughout this experiment? Panels F-H: The legend needs to be revised to explain what is shown in the graphs. Was the statistical significance calculated, and how? Which data points are significantly different?

18) Figure 7A. This is an interesting result because it is hard to explain such robust effect on the DREAM target gene expression by a pretty small difference in the p130 expression levels. This difference is not even apparent in the panel D in this Figure. What is the possible explanation of this result? Also, the endogenous p130 immunoblot should be included in this figure to appreciate the extent of the overexpression (also in figure S4A). Panel D: It appears that only a small fraction of the p130-AA cells undergoes apoptosis. Additional evidence is needed to support the conclusion that this mutant promotes apoptosis.

19) Using DepMap, the authors find that CCNF KO is associated with Palbociclib inactivation of the CDK-RB network, but they fail to show experimentally a correlation between the cyclin F-mediated degradation of p130 and sensitivity to CDK4/6 inhibitors. This should be addressed.

---

## [Author Response]

Essential revisions1) The authors provided evidence that an R680xL682/AxA mutant is unable to interact with cyclin F and that this mutant blocks cell cycle gene expression, cell cycle progression and proliferation. The authors do not acknowledge or account for prior reports that this conserved RxL motif has been implicated in cyclin A and cyclin E binding in both p130 and p107 (Lacy and Whyte, 1997) as well as cyclin D binding in p107 (Leng et al., 2002). Thus, the p130 interaction data and functional data for the AxA mutant are likely to be true and unrelated as the functional data for the AxA likely results from a deficiency in CDK phosphorylation rather than increased stability. The authors need to show that the p130 AxA mutant can still interact with cyclins D, A, and E and be phosphorylated at CDK sites such as S672 for the alternative deficiency in CDK phosphorylation hypothesis to be ruled out for the AxA mutant.

This is a great point. As you point out, an important paper from Lacy and Whyte previously showed that the same sequence motif in p130, which we mapped as being critical for cyclin F binding, stability, ubiquitination, and degradation (amino acids 680-682), could also potentially mediate cyclin E/A binding.

We have addressed this important point with a series of new experiments. First, we showed that the p130(R680A, L682A) mutant (hereafter, p130(AA)) is phosphorylated in vivo at an established CDK site. We performed pulldown experiments with HA-tagged p130(WT) and p130(AA) and blotted them using the published, commercially available, p130 phospho-S672 antibody (new Figure 5 Supplement 2C). This reagent has been well validated by experts in this field (e.g., DeCaprio lab, Litovchick lab, etc.). Both p130(WT) and p130(AA) were similarly phosphorylated on S672. Next, since that antibody only detects p130 phosphorylation at a single site, we also examined in-gel migration of p130. Phosphorylated p130 migrates more slowly in SDS-PAGE. Consistently, HA-tagged versions of p130(WT) and p130(AA) showed similar migration patterns and were similarly impacted by phosphatase treatment post-lysis (new Figure 5 Supplement 2D). Together, these data strongly suggest that p130(AA) is phosphorylated in vivo. Finally, we examined p130 binding to cyclin E and cyclin A by immunoprecipitation (IP). We show that IPs of HA-tagged p130(WT) and p130(AA) copurify cyclin A and cyclin E similarly (See new Figure 5 Supplement 2A). Likewise, IP of endogenous cyclin A co-precipitated indistinguishable levels of HA-tagged p130(WT) and p130(AA) (new Figure 5 Supplement 2B). Altogether, these experiments argue that the impact of the p130(AA) mutant is not the result of a defect in phosphorylation or cyclin E/A binding.

We revisited Lacy and Whyte’s paper, which we should have cited and discussed more clearly. It is notable that they did not analyze phosphorylation of full length p130 in human cells. Rather, they demonstrated that deleting amino acids 680-682 from a fragment of GST-p130 (amino acids 595-675) impaired binding to cyclins in SF9 cell extracts derived from cells overexpressing cyclin A/CDK2 or cyclin E/CDK2 complexes. They showed that this deletion reduced phosphorylation of the GST-p130 fragment, also using insect cell extracts overexpressing cyclin/CDK complexes. They did not test binding of this mutant to cyclins, or its phosphorylation, in cultured human cells.

Also notable are two additional studies, from the Dynlacht and Lukas labs, reporting that cyclin E/A binding to p130 instead relied on a different site in the p130 N-terminus (PMIDs 9710622 and 11157749). And a more recent report showing that p130 phosphorylation by cyclin D-CDK4 depends on a c-terminal helical domain (PMID 30982746). These studies and the Lacy and Whyte paper, have all been cited.

2) If the p130 AxA mutant is competent for CDK phosphorylation, this raises questions regarding the disassembly of the DREAM complex. Previously, CDK4/6 phosphorylation of p130 has implicated in DREAM disassembly but, if the repression of cell cycle gene expression observed in the AxA mutant is due to increased protein stability rather than CDK phosphorylation, this would indicate that p130 degradation is the rate limiting step of DREAM disassembly or that p130 has a transcriptional repression function independent of the DREAM complex. The authors should address p130 degradation within DREAM complex function to resolve this conflict as to reduced p130 degradation leads to increase repression of cell cycle gene expression.

This is an interesting question with respect to whether p130 degradation is limiting, particularly with respect to phosphorylation. Importantly, we found that forced expression of low levels of p130(WT) is sufficient to slow proliferation in NHF cells (Figure 6 Supplement 1C and Figure 6C), consistent with overexpression results from other groups (including the Lacy and Whyte study). Thus, as is the case with RB1, simply increasing the levels of p130 is sufficient to repress proliferation, and this is presumably through its established function in cell cycle gene repression through DREAM.

In our assays, this proliferation defect is profoundly exacerbated by mutation of the cyclin F binding site in p130 (Figure 6), which allows p130 protein to accumulate due to it not being degraded. Our data are consistent with the prior observations where increasing p130 levels in cells is sufficient to slow proliferation. Since the p130(AA) mutant can bind other cyclins and is phosphorylated (see point 1 above), this suggests that degradation is pivotal for p130 inactivation. Further, we determined p130(AA) binding to DREAM, and now show that p130(AA) can assemble into the DREAM complex akin to p130(WT), based on its ability to co-immuno-precipitate with LIN54 (new Figure 5 Supplement 2C). Altogether, this suggests that increasing the level of a phosphorylation proficient p130, by interfering with its destruction, represses cell cycle gene expression and proliferation.

3) Many of the cellular experiments establishing that p130 is cyclin F substrate (Figure 2E, 3, 4B, 5B, S3A) were conducted in 293T cells that express Adenovirus 5 E1A and SV40 Large T antigen which bind and inactivate Rb family members and disrupt the RB-CDK network. Furthermore, SV40 large T promotes dephosphorylation of p130 (Lin and DeCaprio, 2003) explaining why SKP2 overexpression did not alter p130 levels as the SKP2 interacts with phosphorylated p130. SV40 LT also binds directly to FBXW7. Thus, the generalizability of these experiments is limited. Ideally, all the experiments in 293T cells would be repeated in cells where the RB-CDK network was intact but the strong in vitro data for the mapping the cyclin F-p130 interaction on p130 indicates that this may be unnecessary for Figure 2E, 4B, and S3A. At a minimum, the experiments in Figure 3, especially 3B, and 5B should be repeated in cells where the RB-CDK network is intact.

To address this potential concern, we now show that ectopic expression of cyclin F, but not SKP2, bTRCP or Fbxw7, drives p130 degradation in U2OS cells (new Figure 3 Supplement 1A). Further, we now show that cyclin F also regulates p130 half-life in U2OS cells (new Figure 5 Supplement 1A). We have also shown that cyclin F overexpression reduces endogenous p130 protein levels in MCF7 and T47D cells, luminal breast cancer cell lines that do not express large T antigen or HPV oncoproteins (Figure 1D). And, cyclin F depletion increases p130 protein levels in IMR-90 and NHF-1 cell lines, both of which are negative for large T antigen and HPV oncoproteins (Figure 2B).

4) The transition from Figure 1 to Figure 2 is a difficult to follow in the text as Figure 2A and 2B are correlative observations. Figure 2A and 2B feel more like a follow up to 1D than addressing the question of whether and SCF cyclin F complex regulates p130 levels. Consequently, please move Figures 2A and 2B to figure 1 and Figure S2 should be brought into the main figure.

Thank you for these thoughtful suggestions. To accommodate these recommendations, and improve the flow of the manuscript, we have re-organized Figures 1 and 2. Specifically, part of the previous Figure S2, which showed data in NHF-1 cells released from arrest into MLN4924, was moved to Figure 1F. Similarly, the previous Figure 2A, which showed cyclin F and p130 levels in proliferating and quiescent cells across different lines, was moved Figure 1E. The previous Figure 2B, which is similar to the new Figure 1F, was moved to Figure 1 Supplement 3, which also still includes data from T98G cells released from arrest into MLN4924. We have updated the manuscript text accordingly.

5) Figures 6 and 7 are poorly organized and the narrative is a bit difficult to follow and appreciate as the authors bounce between cell cycle, proliferation, and apoptotic readouts. It may strengthen and focus the story to focus on cell cycle gene expression and cell cycle progression in Figure 6 and cellular proliferation and apoptosis in Figure 7. This would also help the reader appreciate the CDK4/6i inhibitor data more as CDK4/6i inhibition is generally considered cytostatic rather than cytotoxic.

We have tried to best accommodate this by making the following changes. Since the apoptotic results are significant, but less well understood, these have been moved to supplemental figures. The phenotype data in Figure 6 begins with showing the protein dynamics of p130(WT) vs p130(AA) and then begins with the broadest of phenotypes, being proliferation. This is followed by cell cycle and 4i imaging of cell cycle related proteins. Then, in Figure 7, we drill down into effects on cell cycle gene expression. We have updated the writing to hopefully make this clearer.

6) Since CDT1 protein species are degraded by multiple complexes, please include RT-qPCR data for CDT1 so the reader may appreciate that the decrease in CDT1 levels results from reduced transcription.

We have included CDT1 RT-qPCR data, demonstrating that the reduction in CDT1 protein levels is due to reduced transcription (see Figures 7A and Figure 7 Supplement 1A).

7) On lines 293-295, the authors compare p130 levels in S4A and 6A to point out the induced p130 AxA accumulates to higher levels at later time points compared to p130 WT. It is difficult to appreciate this comparison as these are different blots in different figures. Consequently, to make this point, the authors should include an IB as in S4A to show p130 WT and AxA levels over the 14 day time course. Ideally, time points would be collected.

This is a great point. We have now included a time course of p130(WT) and p130(AA) levels from zero to 14 days. Levels of p130(AA) accumulate over time (see new Figure 6A).

8) The authors should include representative images for key time points and comparison for the multiple figures of quantitative immunofluorescence experiments, particularly for the nuclear to cytoplasmic ratio. Also the authors should more fully explain their analysis workflow as data can be processed through the "standard modules" cell profiler to differing results. If a previously established workflow was used that has been previously described in the literature, please provide a citation for the workflow.

We have addressed this suggestion by including representative images for phosphorylated RB, CDT1, and CDC6 (Figures 6 and 7). We further elaborated when describing the 4i image acquisition and data processing to clarify methods used.

9) Inactivation of p130 by CDK phosphorylation is likely to dominate p130 degradation in functional assay under normal growth conditions. The authors should consider examining the p130 degradation after DNA damage, in presence of reduced serum levels, or during quiescence (by expression a cyclin F KEN box mutant) to discern the functional impact of altered p130 degradation by p130.

This is a very interesting point. We observed that in several cell lines made quiescent by serum starvation, p130 is upregulated and cyclin F downregulated (Figure 1E). We anticipate that p130 increases in quiescence, at least in part, because of reduced cyclin F. We examined p130 protein levels in response to DNA damage, which has been reported to cause an increase p130 abundance, and during when cyclin F is degraded. While we could recapitulate a p130 increase using MMC at specific doses, as described by others, we see inconsistent results depending on the type of damaging agent applied and with dose of damaging agent. Since these results are inconsistent and inconclusive, we have not included them. We are currently unable to overexpress cyclin F in quiescent cells, precluding that analysis.

Related to this point, and one below, we also now provide new experimental evidence that in NHF-1 cells, p130 is degraded in mitotic NHF cells arrested by nocodazole. Significantly, the p130(AA) mutant which cannot bind or be ubiquitinated by cyclin F, is resistant to degradation in nocodazole arrested cells (see new Figure 5D and Figure 5 Supplement 5C). Examining this further represents an important area of future work but is beyond the scope of the current study.

10) It would help the reader if the results of the iterative immunofluorescence staining were supplemented by additional data such as representative cell images and/or immunoblots. It is difficult to appreciate the extent and the significance of the observed effects in the format shown in Figures 6F-H, 7B-C and S5C-E.

We have addressed this suggestion by including representative images for phosphorylated RB, CDT1, and CDC6 (Figures 6 and 7).

11) Standard methods such as immunoblotting or protein expression in bacteria, are described in excessive detail, whereas the 4i image analysis that could be helpful to a reader unfamiliar with this technique, is described rather briefly. It took a while and reading some additional papers for this reviewer to understand what in fact was shown in the graphs obtained by this technique.

Thank you for this very helpful suggestion. We have addressed this by explaining in better detail the 4i technique and providing representative images of cells from these experiments. We have also reduced other explanations in the manuscript that were described with unnecessary detail.

12) Figure 1. From the description, it appears that results of the DepMap correlation analysis form the premise for this work by showing that depletion of cyclin F has similar impact on the cancer cell fitness as depletion of several components of the Rb-CDK pathway, including p130. The authors then hypothesize that cyclin F may regulate the expression of these factors, which leads to identification of p130 as a cyclin F substrate. Significance of this analysis should be better explained. Indeed, the authors first make the introduction that closely correlated genes in DepMap are often involved in the same protein complex or a functional pathway, and then go on to show that cyclin F and p130 not only have the opposite patterns if expression, but also negatively regulate each other, which would be contradictory to a direct correlation found in DepMap analysis. It would help the reader if the authors explained this apparent contradiction, and ideally validated the conclusions of the DepMap analysis by comparing the effects of cyclin F and p130 knockdown in a panel of select cancer cell lines.

This is an extremely interesting point and we have given this considerable thought. Cyclin D1, CDK4, RBL1 and RBL2, are all highly, positively correlated with cyclin F. We agree that at first glance this is confusing and appears contradictory. However, the more we survey the DepMap dataset, the more it becomes clear that positive and negative correlations cannot always be interpreted as indicating positive and negative relationships between proteins in a pathway.

To be clear, positive and negative correlations often do indicate positive and negative relationships, respectively. For example, the E3 ligase AMBRA1 is negatively correlated with the D-type cyclins, which it targets for degradation. Likewise, CDK4 is positively correlated with Cyclin D1.

However, there are many examples that defy this simple logic. CDK4 is negatively correlated with CDK6, with which it shares a common function. Likewise, D-type cyclins are negatively correlated with each other. These examples are likely explained by the fact that cell lines predominantly use CDK4 or CDK6, or only one D-type cyclin, but not all equally. Nevertheless, these examples demonstrate that caution is needed when interpreting positive and negative DepMap correlations. Furthermore, although Cyclin D1 is indeed positively correlated with CDK4, it is also positively correlated with RBL1 and RBL2, both of which it negatively regulates. Altogether, we think this implies a role for a specific circuitry in a given cell line, while not necessarily implying the direction of the relationship. Simply stated, positive DepMap correlations do not necessarily imply activating or positive enzymatic relationships.

To address your point, we compared fitness scores from the Project Achilles Cancer Dependency map dataset to fitness scores from the Sanger Project Score dataset in ~185 overlapping cell lines utilized by both programs. These represent the two largest and most comprehensive, large scale, CRISPR/Cas9, cancer screening programs. Notably, they are performed independently at different sites, by different individuals, and using different protocols and reagents. Prior analysis showed that these datasets are remarkably well correlated with each other, supporting the reliability of data in both (PMID: 31862961). We also now show correlations between the two datasets for genes in the CDK-RB network in new Figure 1 Supplement 2.

13) Figure 2C: In HeLa cells, all Rb family members are targeted to degradation through the interaction with HPV E7 oncoprotein. Although loss of CCNF appears to increase the levels of p130 in HeLa cells, this result does not strongly support the direct role of CCNF in p130 degradation without ruling out an indirect effect on E7 function. Additionally, loss of CCNF could influence the cell cycle synchronization therefore the cell cycle analysis data should be included with this experiment. In general, since cyclin F is involved in regulation of many cell-cycle related factors, the cell cycle data should be included in the experiments where the levels of CCNF are manipulated such as Figure 2D.

We have addressed this potential concern in several ways. First, we provide cell cycle flow cytometry data from cells in cyclin F siRNA-mediated depletion experiments (Figure 2 Supplement 1B-C) as well as for asynchronously proliferating cyclin F KO and control HeLa cells (Figure 2 supplement 1D). Consistently, we and others previously showed that the cyclin F KO HeLa cells are well adapted and grow quite normally in culture (similarly to knockout of most canonical cyclins). In addition, the HeLa KO synchronization experiment was blotted for cyclin E, which shows very similar kinetics between control KO and cyclin F KO cell lines (see Figure 2 Supplement 1A). Since cyclin F is the only thing different between cell lines, this strongly argues that even if E7 were involved in this cell line, that this still is a cyclin F dependent effect.

14) Figure 2E: Again, the choice of 293T cell line to demonstrate the binding between the endogenous p130 and cyclin F is surprising. The levels of p130 in these cells are very low because of the presence of SV40 Large T antigen that targets Rb family members for degradation. Furthermore, SV40 LT binds to Rb family members and can indirectly recruit other factors into the complexes with these proteins. Therefore, the endogenous interaction between p130 and cyclin F should be confirmed in another cell line. Figure 3: Same concerns – the presence of SV40 LT could complicate the interpretation of these results. The authors need to confirm these effects in another cell line, in the absence of SV40 LT.

We addressed this potential concern by performing an endogenous IP for cyclin F in U2OS cells, which showed that these endogenous proteins interact in this cell line as well (see new Figure 2C). In addition, we also now show that cyclin F expression can cause the degradation of p130 in U2OS cells, but that SKP2, bTRCP and Fbxw7 cannot (Figure 3 Supplement 1A).

15) Figure 4: Panel B: It would be interesting to know whether R680A/L682A p130 mutant can still bind SV40 Large T or HPV E7 proteins. Panel E: The rationale for using E2F1 displacement for measuring the binding affinity between cyclin F-Skp1 complex and p130 fragments is unclear. Is it possible to show a direct binding between these factors?

We have addressed this suggestion by analyzing binding of HA-tagged versions of p130(WT) and p130(AA) to E7 in HeLa cells and large T antigen in HEK293T cells. We see no difference in binding in either of these situations (see new Figure 5 Supplement 2E-F).

We also re-analyzed cyclin F IP-MS data from 293T cells for the presence of large T antigen and found that we did not enrich for large T antigen in those experiments. These data are not yet published.

16) Figure 5B: It is apparent that p130-AA is still being degraded during the CHX chase, but is unaffected by cyclin F overexpression. Could the proteasome or neddylation inhibitors block this degradation? Is it possible that both Skp2 and cyclin F can degrade p130 through different mechanisms? Panel C: Significance of the differences is not shown.

It is possible that Skp2 contributes to p130 degradation, and it would be difficult to rule this out completely. The focus of our paper is not to discern the potential role of Skp2 in p130 degradation. We tested Skp2 by overexpression in two cell lines, one of which was used in the original studies reporting a role for Skp2 (HEK293T cells) and obtained two negative results. We are concerned about overinterpretation of the CHX data, as this results in complete inactivation of all translation, and causes significant cellular stress. If MLN4924 completely blocked degradation in that assay, it still would not imply a role for Skp2 versus the hundreds of other cullin E3s.

17) Figure 6: It would be very interesting to see how the p130-AA levels are regulated during the G0/G1 arrest and progression through the cell cycle. This could be a key experiment to demonstrate the role of the 680-RRL site for cell cycle dependent degradation of p130. Panel E: Description of this figure indicates that cells were pulse-labeled with EdU to detect S-phase, but the figure only shows G0/G1 fraction. Changes in S-phase and G2/M fractions could be informative and should be shown. Also, this experiment needs to be better explained. Were the cells allowed to grow to confluency over 14 days? Was the growth media changed or supplemented throughout this experiment? Panels F-H: The legend needs to be revised to explain what is shown in the graphs. Was the statistical significance calculated, and how? Which data points are significantly different?

These are helpful suggestions. We had originally shown only G0/G1 cells in Figure 6 for simplicity but have now added the remaining data to the Figure 6 Supplement 1D.

The cells were not grown to confluency at any point in our proliferation assays, but rather were split/fed regularly, and this is now noted in the text. The data in panels is more clearly explained in the legend, and they have been updated to reflect statistical significance.

To address your first point, we synchronized cells expressing HA-tagged versions of p130(WT) and p130(AA) in different phases of the cell cycle. While we observed little or no difference in most conditions, to our surprise, we found that p130(WT) is degraded in nocodazole arrested NHF cells (see new Figure 5D). Likewise, endogenous p130 protein is degraded in nocodazole arrested NHF cells (see new Figure 5 Supplement 1C). Significantly, we did not observe p130(AA) degradation in nocodazole-synchronized cells (Figure 6E).

18) Figure 7A. This is an interesting result because it is hard to explain such robust effect on the DREAM target gene expression by a pretty small difference in the p130 expression levels. This difference is not even apparent in the panel D in this Figure. What is the possible explanation of this result? Also, the endogenous p130 immunoblot should be included in this figure to appreciate the extent of the overexpression (also in figure S4A). Panel D: It appears that only a small fraction of the p130-AA cells undergoes apoptosis. Additional evidence is needed to support the conclusion that this mutant promotes apoptosis.

To address the point regarding apoptosis we have now carried out annexin V staining and flow cytometry, and we observe the same effect. The increase in apoptosis is clear, reproducible, and statistically significant. We had originally analyzed apoptosis after seeing the dramatic impact on cell growth/fitness, and to rule out a role for apoptosis. Defying our hypothesis, our data indicated that a small amount of apoptosis is indeed occurring. These new flow cytometry data are included in Figure 7 Supplement 2B.

Since p130(AA) can assemble into the DREAM complex, and the protein can still be phosphorylated, the data in Figure 7A argues strongly that even a small increase in p130 expression is sufficient to impair cell cycle progression. Since there is likely little p130 in cycling cells, blocking its ubiquitination is sufficient to cause a strong defect in cell cycle gene expression programs and proliferation.

19) Using DepMap, the authors find that CCNF KO is associated with Palbociclib inactivation of the CDK-RB network, but they fail to show experimentally a correlation between the cyclin F-mediated degradation of p130 and sensitivity to CDK4/6 inhibitors. This should be addressed.

The DepMap consortium utilized barcoding strategies in hundreds of cells lines to determine sensitivities to myriad drugs, in an effort at drug repurposing. Our results show that cyclin F regulates p130. The DepMap analysis is consistent with our results, in that cyclin F has an impact on cells that is highly similar to CDK4, cyclin D, and also Palbociclib, all of which are negative regulators of the RB-family of proteins, and whose inactivation can enhance resistance to Palbociclib. Determining chemical-genetic interactions between CDK4/6 inhibitors and cyclin F is an important area of future study, but at this time, beyond the scope of this current study. We have therefore moved this observation to the Discussion, following consultation with the editor, and discussed it there appropriately.